# Continual Diffusion: Continual Customization of Text-to-Image Diffusion with C-LoRA

**James Seale Smith**                                              *jamessealesmith@gatech.edu*
*Samsung Research America*
*Georgia Institute of Technology*

**Yen-Chang Hsu**
*Samsung Research America*

**Lingyu Zhang**
*Samsung Research America*

**Ting Hua**
*Samsung Research America*

**Zsolt Kira**
*Georgia Institute of Technology*

**Yilin Shen**
*Samsung Research America*

**Hongxia Jin**
*Samsung Research America*

**Reviewed on OpenReview:** *https://openreview.net/forum?id=TZdEgwZ6f3*

## Abstract

Recent works demonstrate a remarkable ability to customize text-to-image diffusion models while only providing a few example images. What happens if you try to customize such models using multiple, fine-grained concepts in a sequential (i.e., continual) manner? In our work, we show that recent state-of-the-art customization of text-to-image models suffer from catastrophic forgetting when new concepts arrive sequentially. Specifically, when adding a new concept, the ability to generate high quality images of past, similar concepts degrade. To circumvent this forgetting, we propose a new method, C-LoRA, composed of a continually self-regularized low-rank adaptation in cross attention layers of the popular Stable Diffusion model. Furthermore, we use customization prompts which do not include the word of the customized object (i.e., "person" for a human face dataset) and are initialized as completely random embeddings. Importantly, our method induces only marginal additional parameter costs and requires no storage of user data for replay. We show that C-LoRA not only outperforms several baselines for our proposed setting of text-to-image continual customization, which we refer to as Continual Diffusion, but that we achieve a new state-of-the-art in the well-established rehearsal-free continual learning setting for image classification. The high achieving performance of C-LoRA in two separate domains positions it as a compelling solution for a wide range of applications, and we believe it has significant potential for practical impact.

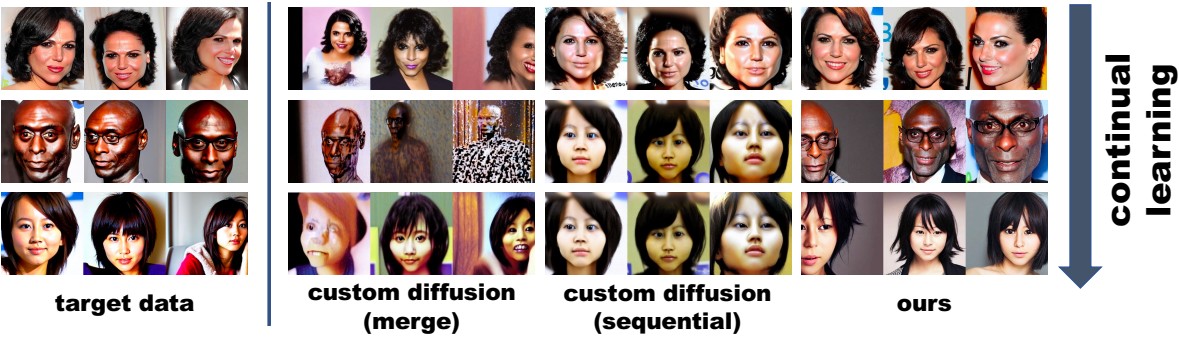

Figure 1: We learn to generate text-conditioned images of new concepts in a sequential manner (i.e., **continual learning**). Here we show three concepts from the learning sequence sampled *after* training **ten concepts sequentially**. SOTA Custom Diffusion (Kumari et al., 2022) suffers from catastrophic forgetting, so we propose a new method which drastically reduces this forgetting.

# 1    Introduction

Text-to-image generation is a rapidly growing research area that aims to develop models capable of synthesizing high-quality images from textual descriptions. These models have the potential to enable a wide range of applications, such as generating realistic product images for e-commerce websites, creating personalized avatars for virtual reality environments, and aiding in artistic and creative endeavors. Recent advances in this area have shown significant progress in generating images with fine-grain details that can be customized to specific concepts, such as generating a portrait of one's self or pet, while providing only a few example images to "instruct" the model. We ask a simple and practical question: *What happens when these concepts arrive in a sequential manner?*

Specifically, we advocate that one of the main challenges in these models is to *continuously* customize them with new, previously unseen fine-grained concepts, while only providing a few examples of these new concepts. **We refer to this new setting as Continual Diffusion**, and to the best of our knowledge we are the first to propose this problem setting. Continual Diffusion provides a more efficient engagement with the text-to-image tools, such as visual storytelling and content creation. A key aspect is to not require the model to be re-trained on past concepts after each new concept is added, alleviating both computational and data privacy burdens. However, this is also a *challenging problem* as we show it induces strong interference and catastrophic forgetting in existing methods.

In this paper, we start by analyzing existing customized diffusion methods in the popular Stable Diffusion model (Rombach et al., 2022), showing that these models catastrophically fail for sequentially arriving fine-grained concepts (we specifically use human faces and landmarks). Motivated by the recent CustomDiffusion (Kumari et al., 2022) method, as well as recent advances in continual learning, we then propose a *continually self-regularized low-rank adaptation* in cross attention layers, which we refer to as **C-LoRA**. Our method leverages the inherent structure of cross attention layers and adapts them to new concepts in a low-rank manner, while simultaneously preserving the knowledge of past concepts through a self-regularization mechanism. Additionally, we propose a custom tokenization strategy that (i) removes the word of the customized object (typically used in prior works) and (ii) is initialized as random embeddings (rather than arbitrary embeddings of lesser-used words, as done previously), and show that this outperforms commonly used customization prompts.

As discussed in Figure 2, we note that existing approaches such as learning task-specific adapters or customized tokens/words (Gal et al., 2022) are not satisfying solutions. For the former, this type of approach would create different models per concept, inhibiting the ability to generate content containing *multiple learned concepts*. For the latter, this type of approach underperforms model fine-tuning when capturing fine-grained concepts such as *your* specific face instead of faces in general. Thus, it is critical to find a

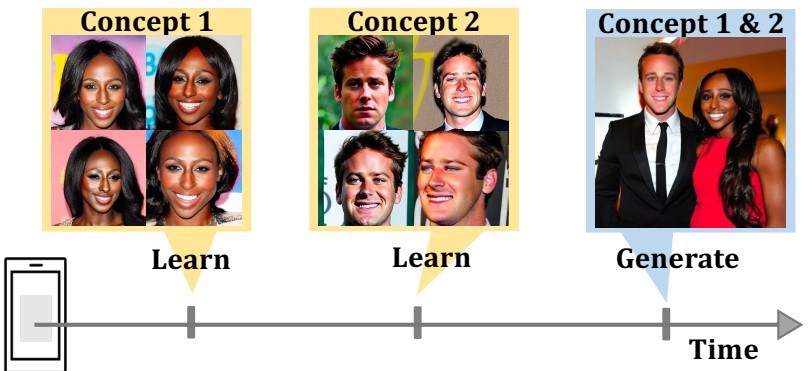

Figure 2: A use case of our work - a mobile app sequentially learns new customized concepts. At a later time, the user can generate photos of prior learned concepts. The user should be able to generate photos with *multiple concepts together*, thus ruling out simple methods such as per-concept adapters. Furthermore, the concepts are *fine-grained* (e.g., human faces), and simply learning new tokens or words as done in Textual Inversion (Gal et al., 2022) is not effective.

solution like ours which gently adapts the model to learn these new concepts *without interfering with or forgetting prior seen concepts*. **In summary, our contributions in this paper are fourfold:**

1. We propose the setting of continual customization of text-to-image models, or simply Continual Diffusion, and show that current approaches suffer from catastrophic forgetting in this setting.
2. We propose C-LoRA, which efficiently adapts to new concepts while preserving the knowledge of past concepts through continual, self-regularized, low-rank adaptation in cross attention layers of Stable Diffusion.
3. We show that our C-LoRA alleviates catastrophic forgetting in the Continual Diffusion setting with both quantitative and qualitative analysis.
4. We extend C-LoRA into the well-established setting of continual learning for image classification, and demonstrate that our gains translate to state-of-the-art performance in the widely used benchmark, demonstrating the versatility of our method.

## 2 Background and Related Work

**Conditional Image Generation Models**: Conditional image generation is an active research area, with notable works including Generative Adversarial Networks (GANs) (Goodfellow et al., 2014; Karras et al., 2020), Variational Autoencoders (VAE) (Kingma & Welling, 2013), and more recently, diffusion models (Dhariwal & Nichol, 2021; Ho et al., 2020; Sohl-Dickstein et al., 2015). As GAN-based models require delicate parameter selection and control over generation results usually comes from pre-defined class labels (Mirza & Osindero, 2014), we focus on diffusion-based conditional image generation that takes free-form text prompts as conditions (Ramesh et al., 2022; Liu et al., 2022a). Specifically, diffusion models operate by learning a forward pass to iteratively add noise to the original image, and a backward pass to remove the noise for a final generated image. A cross attention mechanism is introduced in the (transformer-based) U-Net (Ronneberger et al., 2015) to inject text conditions.

Recent works have gone beyond text-only guidance to generate *custom* concepts such as a unique stuffed toy or a pet. For example, Dreambooth (Ruiz et al., 2022) fine-tunes the whole parameter set in a diffusion model using the given images of the new concept, whereas Textual Inversion (Gal et al., 2022) learns only custom feature embedding "words". While these approaches are used for customizing to a single concept, Custom Diffusion (Kumari et al., 2022) learns *multiple concepts* using a combination of cross-attention fine-tuning, regularization, and closed-form weight merging. However, we show this approach struggles to learn similar, fine-grained concepts in a *sequential* manner (i.e., Continual Diffusion), thus motivating the

need for our work. We note that a concurrent work, ZipLoRA (Shah et al., 2023), bears resemblance to our approach; we compare to this method in Section 4.

**Continual Learning**: Continual learning is the setting of training a model on a sequence of tasks, where each task has a different data distribution, without forgetting previously learned knowledge. Existing methods can be broadly classified into three categories based on their approach to mitigating catastrophic forgetting (McCloskey & Cohen, 1989). Regularization-based methods (Douillard et al., 2020; Kirkpatrick et al., 2017; Li & Hoiem, 2017; Zenke et al., 2017) add extra regularization terms to the objective function while learning a new task. For example, EWC (Kirkpatrick et al., 2017) estimates the importance of parameters and applies per-parameter weight decay. Rehearsal-based methods (Chaudhry et al., 2019a;b; Hou et al., 2019; Kamra et al., 2017; Ostapenko et al., 2019; Pham et al., 2021; Rebuffi et al., 2017; Rolnick et al., 2019; Smith et al., 2021; van de Ven et al., 2020) store or generate samples from previous tasks in a data buffer and replay them with new task data. However, this approach may not always be feasible due to privacy or copyright concerns. Architecture-based methods (Aljundi et al., 2017; Li et al., 2019; Rusu et al., 2016; Yoon et al., 2017) isolate model parameters for each task. Recently, prompt-based continual learning methods for Vision Transformers such as L2P (Wang et al., 2022c), DualPrompt (Wang et al., 2022b), S-Prompt (Wang et al., 2022a), and CODA-Prompt (Smith et al., 2022b) have outperformed rehearsal-based methods without requiring a replay buffer. These methods work well for classification type problems, but it is unclear how they would be applied to text-to-image generation given that their contributions focus on inferring discriminative properties of the data to form prompts for classification. However, we do compare to these methods in their original setting. We note that generative continual learning is not new (Lesort et al., 2019; Shin et al., 2017; Zhai et al., 2019), but emphasize that these are train-data hungry methods (whereas ours is few-shot fine-tuning), and they do *not* operate on fine-grained tasks such as individual faces.

While the aforementioned work is mainly on the uni-modal continual learning, a few recent approaches have been proposed for the multimodal setting. REMIND (Hayes et al., 2020) proposed continual VQA tasks with latent replay, but their method requires storing compressed training data. CLiMB (Srinivasan et al., 2022) proposed CL adaptation to coarsely different VL tasks, such as Visual Question Answering(VQA), Natural Language for Visual Reasoning (NLVR), Visual Entailment (VE), and Visual Commonsense Reasoning(VCR), assuming knowledge of the evaluated task-id at inference time. Construct-VL (Smith et al., 2022a) focuses on the task of natural language visual reasoning. We formulate our problem as continual adaptation of Stable Diffusion to multiple, fine-grained concepts, and the majority of the methods discussed in this section do not have a one-to-one translation for our problem setting.

**Parameter-Efficient Fine-Tuning**: There have been several proposed approaches for fine-tuning models with fewer parameters, including adapters, low-rank adapters, prompt learning, or simply fine-tuning subsets of the model. Houlsby et al. (2019) introduced one of the earliest frameworks for parameter-efficient transfer learning in NLP, proposing the use of adapters to fine-tune pre-trained models efficiently. Building on these ideas, subsequent advancements have included methods like prefix tuning(Li & Liang, 2021) and prompt tuning (Lester et al., 2021), which further refine the process of adapting large models to specific tasks with minimal parameter updates. Some recent works in this area include Delta-T5 (Ding et al., 2022), UNIPELT (Mao et al., 2021), Polyhistor (Liu et al., 2022b), and VL-ADAPTER (Sung et al., 2022). One promising approach is to use low-rank adapters, such as LoRA (Hu et al., 2021), which have been shown to be effective in adapting models to new tasks with minimal additional parameters. In this work, we build upon this concept and propose an efficient continual learning neural architecture that layers low-rank adapters on the key and value projection matrices of Stable Diffusion.

## 3 Method

In our Continual Diffusion setting, we learn $N$ customization "tasks" $t \in \{1, 2, \ldots, N-1, N\}$, where $N$ is the total number of concepts that will be shown to our model. Inspired by Custom Diffusion (Kumari et al., 2022), the high level intuition of our method is to find a *new and continuous* way to update a small number of weights in Stable Diffusion. Our observation is that we should both 1) update our model in a way that does not over-fit to our new concept and "leave room" to add future concepts, and 2) not overwrite the information learned from the past concepts (i.e., avoid *catastrophic forgetting*). We re-emphasize that we

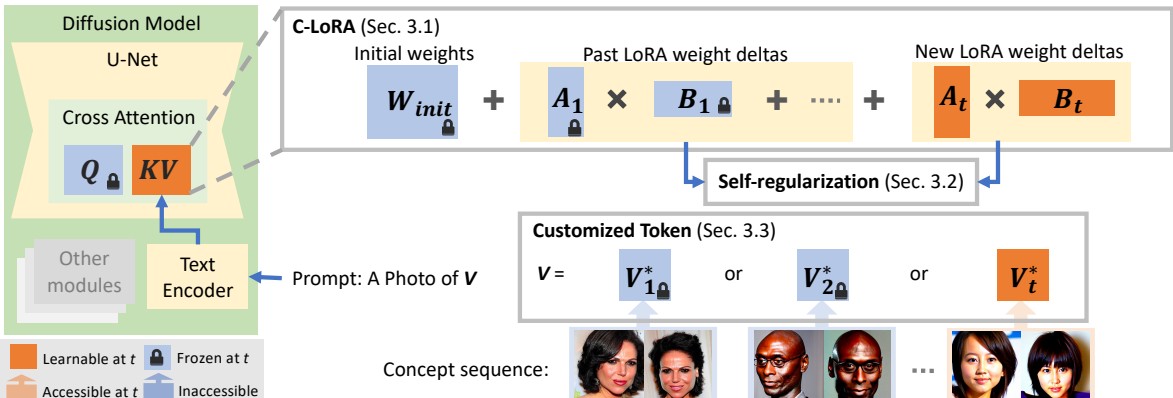

Figure 3: Our method, C-LoRA, updates the key-value (K-V) projection in U-Net cross-attention modules of Stable Diffusion using a continual, self-regulating low-rank weight adaptation. The past LoRA weight deltas are used to regulate the new LoRA weight deltas by guiding which parameters are most available to be updated. Unlike prior work (Kumari et al., 2022), we initialize custom token embeddings as random features and remove the concept name (e.g., "person") from the prompt.

cannot simply use single-concept adapters, because we desire a model which can produce images of multiple learned concepts at the same time. For example, a user of this method may want to produce a photo with a grandparent sitting with the user's new pet in a novel setting - single adapters per concept would not enable such a functionality.

Thus, we propose a method with contributions that are two-fold. First, we fine-tune a small number of weights using continual, self-regularized low-rank adaptors (LoRA (Hu et al., 2021)), which we refer to as *C-LoRA*. We build on LoRA due to (1) its parameter efficiency (allowing us to store past task parameters for regularization of future tasks), (2) inference efficiency (the learned parameters can be folded back into the original model weights, and thus has zero cost to inference), (3) its unique ability to self-regularize (proposed by us), which cannot be translated to other parameter-efficient fine-tuning methods, and (4) prior success in VL discriminate tasks[1] (Smith et al., 2022a).

Our C-LoRA contains two key ingredients: (1) we add new LoRA parameters per task, and (2) we use past LoRA parameters to guide our new parameters to adapt the weight matrix in parameters which are less likely to interfere. We keep personalized token embeddings far from each other in the token embedding space with random initialization and remove any object names (different from prior works). This encourages the personalized tokens to give unique "personalized instructions" to the updated cross-attention layers, rather than having overlapping instructions which can interfere with past and future concepts. The remainder of this section describes the fine details of our approach.

## 3.1 C-LoRA

Following the findings of Custom Diffusion (Kumari et al., 2022), we only modify a small number of parameters in the Stable Diffusion model. *Why?* In addition to its described empirical success, this is intuitively inviting for *continual learning* in that we reduce the spots in the model which can suffer from catastrophic forgetting by surgically modifying highly sensitive parameters. However, rather than fine-tune these parameters (as is done in Custom Diffusion), which would induce favorable conditions for interference and catastrophic forgetting, we propose low-rank adaptations designed to self-regularize to avoid the aforementioned issues.

*Which parameters do we modify?* Stable Diffusion contains three parts: the variational autoencoder (VAE), denoising model U-Net, and the text encoder. Kumari et al. (2022) show that the cross-attention parameters

---

[1]This work focuses on using LoRA to create pseudo-rehearsal data from past tasks for a discriminative task; this is not applicable to a generation problem as we already have generated data, however we do compare to the "base" part of their method in our experiments: LoRA (sequential).

of the U-Net are most sensitive to change during Stable Diffusion customization, and therefore propose to modify only these blocks. We modify the same module, but with a different approach.

Consider the single-head cross-attention (Vaswani et al., 2017) operation given $\mathcal{F}_{attn}(Q, K, V) = \sigma\left(\frac{QK^\top}{\sqrt{d'}}\right)V$ where $\sigma$ is the softmax operator, $Q = \boldsymbol{W}^Q \boldsymbol{f}$ are query features, $K = \boldsymbol{W}^K \boldsymbol{c}$ are key features, $V = \boldsymbol{W}^V \boldsymbol{c}$ are value features, $\boldsymbol{f}$ are latent image features, $\boldsymbol{c}$ are text features, and $d'$ is the output dimension. The matrices $\boldsymbol{W}^Q, \boldsymbol{W}^K, \boldsymbol{W}^V$ map inputs $\boldsymbol{f}$ and $\boldsymbol{c}$ to the query, key, and value features, respectively.

Following Kumari et al. (2022), we only modify $\boldsymbol{W}^K, \boldsymbol{W}^V$, which project the text features, and we refer to these as simply $\boldsymbol{W}^{K,V}$. When learning customization "task" $t$, we minimize the following loss:

$$\min_{\boldsymbol{W}_t^{K,V} \in \theta} \quad L_{diff}(x, \boldsymbol{\theta}) + \lambda \mathcal{L}_{forget}(\boldsymbol{W}_{t-1}^{K,V}, \boldsymbol{W}_t^{K,V}) \tag{1}$$

where $x$ is the input data of the new concept, $\mathcal{L}_{diff}$ is the loss function for stable diffusion w.r.t. model $\theta$ (we do not modify this in our work), $\mathcal{L}_{forget}$ minimizes forgetting between old task $\boldsymbol{W}_{t-1}^{K,V}$ and new task $\boldsymbol{W}_t^{K,V}$, and $\lambda$ is a hyperparameter chosen with a simple exponential sweep.

As previously mentioned, we parameterize the weight delta between old task $\boldsymbol{W}_{t-1}^{K,V}$ and new task $\boldsymbol{W}_t^{K,V}$ using LoRA (Hu et al., 2021) parameters, which decomposes the weight matrices into low-rank residuals, or:

$$\begin{aligned}
\boldsymbol{W}_t^{K,V} &= \boldsymbol{W}_{t-1}^{K,V} + \boldsymbol{A}_t^{K,V} \boldsymbol{B}_t^{K,V} \\
&= \boldsymbol{W}_{init}^{K,V} + \left[\sum_{t'=1}^{t-1} \boldsymbol{A}_{t'}^{K,V} \boldsymbol{B}_{t'}^{K,V}\right] + \boldsymbol{A}_t^{K,V} \boldsymbol{B}_t^{K,V}
\end{aligned} \tag{2}$$

where $\boldsymbol{A}_t^{K,V} \in \mathbb{R}^{D_1 \times r}$, $\boldsymbol{B}_t^{K,V} \in \mathbb{R}^{r \times D_2}$, $\boldsymbol{W}^{K,V} \in \mathbb{R}^{D_1 \times D_2}$, and $r$ is a hyper-parameter controlling the rank of the weight matrix update, chosen with a simple grid search. $\boldsymbol{W}_{init}^{K,V}$ are the initial values from the pre-trained model. This parameterization creates the surface to inject our novel regularization, described next.

### 3.2 Self Regularization

*How do we prevent forgetting?* Prior experience would first suggest to provide simple replay data (real or generated) for knowledge distillation (Lesort et al., 2019; Shin et al., 2017). However, as we show later in this paper, generative replay induces new artifacts and catastrophic forgetting, and additionally we want to avoid storing all training data for data privacy and memory efficiency. Another approach is to estimate individual parameter importance for each task, and penalize changes to these parameters (Aljundi et al., 2018; Kirkpatrick et al., 2017). However, such a strategy stores a copy of original parameters, introducing significant overhead, and additionally *we directly show in our experiments that C-LoRA has superior performance.*

Instead, we devise a simple regularization that works well without storing replay data and introduces zero parameters beyond LoRA parameters (Eq. 2). The high level idea is that we penalize the LoRA parameters $\boldsymbol{A}_t^{K,V}$ and $\boldsymbol{B}_t^{K,V}$ for altering spots that have been edited by previous concepts in the corresponding $\boldsymbol{W}_t^{K,V}$. Thus, we use the summed products of past LoRA parameters themselves to penalize the future changes. Specifically, we regularize using:

$$\mathcal{L}_{forget} = \left|\left|\left|\sum_{t'=1}^{t-1} \boldsymbol{A}_{t'}^{K,V} \boldsymbol{B}_{t'}^{K,V}\right| \odot \boldsymbol{A}_t^{K,V} \boldsymbol{B}_t^{K,V}\right|\right|_F^2 \tag{3}$$

where $\odot$ is the element-wise product (also known as the Hadamard product), $|\cdot|$ is the element wise absolute value, and $||\cdot||_F$ denotes the Frobenius norm. This simple and intuitive penalty "self-regularizes" future LoRA with high effectiveness and efficiency. We highlight that the $\boldsymbol{A}$ and $\boldsymbol{B}$ are *low-rank matrices* and thus only incur small costs to training and storage.

### 3.3 Customized Token Strategy

To further mitigate interference between concepts, we propose a new custom tokenization strategy. Specifically, we add $N$ personalized tokens $V_1^*, V_2^*, \dots, V_N^*$ to the input sequence, where $N$ is the total number of

concepts the model will learn. Unlike Custom Diffusion (Kumari et al., 2022), which initializes these tokens from lesser-used words, we initialize with random embeddings in the token embedding space. Furthermore, we remove any object names from the input sequence to encourage unique personalized embeddings. During inference, we replace the personalized tokens with the learned embeddings of the corresponding concept, allowing the model to produce images of learned concepts at the same time.

We found that this tokenization approach improves the model's ability to learn multiple concepts without interference. *Why is it better?* When including the object name (e.g., "person"), or using this word for token initialization, we found interference to be much higher. For example, many concepts are completely "overwritten" with future task data. When initializing with random, lesser-used tokens, we found that some initializations were better than others, leading the "good" tokens to incur less forgetting than the "bad" tokens, and thus remaining unsatisfactory and motivates our strategy.

**Putting it Together.** Our final approach, summarized in Figure 3, introduces a simple continual custom tokenization along with our self-regularized, low-rank weight adaptors (C-LoRA). Our method is parameter efficient and effective, as demonstrated in *two different* continual learning experiment settings in the following sections.

## 4 Continual Text-To-Image Experiments

**Implementation Details.** For the most part, we use the same implementation details as Custom Diffusion (Kumari et al., 2022) with 2000 training iterations, which is the best configuration for generating faces for baseline methods. To improve the fairness of our comparison with Custom Diffusion, we regularizes training with auxiliary data (as done in Custom Diffusion) for *all* methods. However, using real images would require constant access to internet downloads, which would be highly impractical for our setting. Thus, in the spirit of "continual learning", we instead generate[2] regularization images from the Stable Diffusion backbone, as done in one of the Custom Diffusion variants, using the prompt "a photo of a person". We tuned hyperparameters on a short task sequence (5) to avoid over-fitting to the full task sequence (van de Ven & Tolias, 2019). For LoRA, we searched for the rank using a simple exponential sweep and found that a rank of 16 sufficiently learns all concept. Additional training details are located in Appendix C.

**Metrics.** Unlike qualitative comparisons for generative models, where performance has plateaued and fine-grained visual differences across methods are hard to spot, our method starkly increases image generation quality during Continual Diffusion compared to competing methods. Most methods either produce images with severe artifacts *or* catastrophically erase past concepts and produce images with the wrong identity. However, we do provide some quantitative results calculating the similarity between training data samples and diffusion model samples. Given two batches of data samples, we first embed both into a rich semantic space using CLIP (Radford et al., 2021) image embedding, and then we calculate the distribution distance using the Maximum Mean Discrepancy (MMD) (Gretton et al., 2012) with a polynomial kernel. The difference between our metric and the image similarity metric from Kumari et al. (2022) is that we use a true distribution distance (MMD) instead of 1-to-1 cosine similarity calculations; however, we also show their CLIP image-alignment metric in our main tables and the remaining metrics in our Appendix. Specifically, we provide CLIP image-alignment (Gal et al., 2022) in our main text and Appendix, and both CLIP text-alignment (Hessel et al., 2021) and KID (Bińkowski et al., 2018) in Appendix A only.

From this score, we present the mean MMD score *averaged over all concepts and measured after training the final concept* as $A_{mmd}$, where lower is better. We also report the average change in past concept generations from task to task as a "forgetting" metric, given as $F_{mmd}$, where lower is also better. The exact formulas for the metrics are found in Appendix A. We note that $A_{mmd}$ is the more important metric, and $F_{mmd}$ simply provides more context (e.g., did the method learn the concept well and forget, as we see with Custom Diffusion, or did the method not capture the concept well to start with, as we see with Textual Inversion?). We additionally report both the number of parameters *trained* (i.e., unlocked during training a task) and

---

[2]Kumari et al. (2022) show that using generated images instead of real images leads to similar performance on the target concepts.

*stored* (i.e., stored between tasks). We discuss this more in Appendix C. These are reported in % of the U-Net backbone model for easy comparison.

**Baselines.** We compare to recent customization methods Textual Inversion (Gal et al., 2022), Dream-Booth (Ruiz et al., 2022), and Custom Diffusion (Kumari et al., 2022). For Custom Diffusion, we compare to both sequential training (denoted as *sequential*) as well as the constrained merging optimization variant (denoted as *merged*) which stores separate KV parameters for each concept individually and then merges the weights together into a single model using a closed-form optimization (see Kumari et al. (2022) for more details). We also compare to continual learning methods Generative Replay (Shin et al., 2017) and EWC³ (Kirkpatrick et al., 2017), which we have both engineered to work with Custom Diffusion. We further contextualize our results by drawing a comparison with a concurrent work, ZipLoRA (Shah et al., 2023). We developed an implementation of ZipLoRA from the ground up to adapt it to our continual learning framework. ZipLoRA does not perform as well as our C-LoRA, likely due to ZipLoRA being originally conceived for concept/style integration, as opposed to learning a large number of subjects in a continual learning context. Finally, we ablate our method by comparing to a continual variant of LoRA (Hu et al., 2021; Smith et al., 2022a), which we refer to as LoRA (sequential). Additional ablations, including for our tokenization strategy, are located in Appendix B.

### 4.1 Continual Customization of Real Faces

We first benchmark our method using the 512x512 resolution (self-generated) celebrity faces dataset, Celeb-Faces Attributes (Celeb-A) HQ (Karras et al., 2017; Liu et al., 2015). We sample 10 celebrities at random which have at least 15 individual training images each. Each celebrity customization is considered a "task", and the tasks are shown to the model sequentially. Qualitative results are shown in Figure 4 showing samples from task 1, 6, and 10 *after training all 10 tasks*, while quantitative results are given in Table 1. Here, we see quite clearly that our method has tremendous gains over the existing techniques. [a] and [b] give the target data and *offline* (i.e., **not** our setting) Custom Diffusion, respectively. [c] Textual Inversion (Gal et al., 2022) struggles to capture the dataset individuals, though it suffers no forgetting (given the backbone is completely frozen). [e] Deep Generative Replay (Shin et al., 2017) has completely collapsed - we noticed that artifacts begin to appear in early tasks, and then as the model is trained on these artifacts in future tasks, the collapsing performance results in a "snowball" effect. [d] DreamBooth (Ruiz et al., 2022) and [f] Custom Diffusion (Kumari et al., 2022) (sequential) are prone to high catastrophic forgetting, with the former suffering more (which is intuitive given DreamBooth trains the full model). Notice that the first two rows for DreamBooth and middle row for Custom Diffusion (sequential) are almost identical to their respective third rows, *and more importantly are the completely wrong person.* While we noticed [g] Custom Diffusion (merge) performed decent in the short 5 task sequence wrt $A_{mmd}$ (2.62 compared to our 1.65, yet still under-performs our method), it fails to merge 10 tasks (8.89 compared to our 2.04), evident by the strong artifacts present in the generated images. [h] EWC (Kirkpatrick et al., 2017) has less forgetting, but we found that turning up the strength of the regularization loss prohibited learning new concepts, yet relaxing the regularization led to more forgetting. We notice that [i] LoRA (sequential) (Hu et al., 2021; Smith et al., 2022a) offers small improvements over sequential Custom Diffusion, yet still has high forgetting. Finally, we note in Table 1 that the concurrent ZipLoRA (Shah et al., 2023) performs the highest of all competitor methods, but it does not perform as well as our C-LoRA.

On the other hand, we see that [j] our method *produces recognizable images for all individuals*, and furthermore has the best $A_{mmd}$ performance. Notice that our method adds significantly fewer parameters than Custom Diffusion Merge (Kumari et al., 2022) and has clear superior performance.

### 4.2 Continual Customization of Landmarks

As an additional dataset, we demonstrate the generality of our method and introduce an additional dataset with a different domain, benchmarking on a 10 length task sequence using the Google Landmarks dataset v2 (Weyand et al., 2020), which contains landmarks of various resolutions. Specifically, we sampled North

---

³EWC is tuned in the same way as our method's regularization loss (simple exponential sweep) and we report the run with the best $A_{mmd}$.

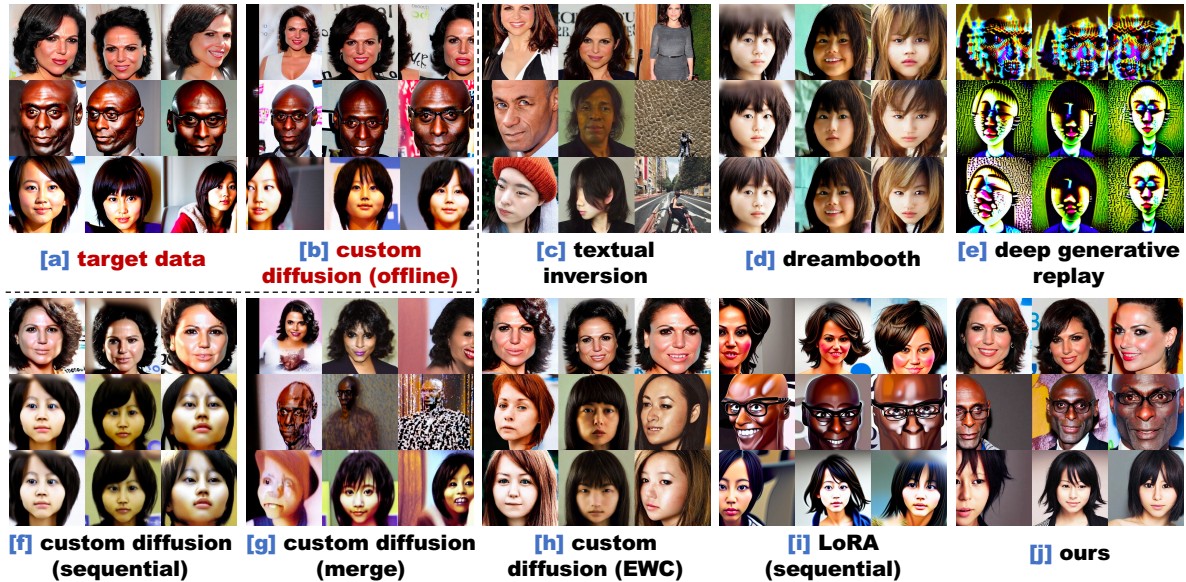

Figure 4: Qualitative results of continual customization using the Celeb-A HQ (Karras et al., 2017; Liu et al., 2015) dataset. Results are shown for concepts (tasks) 1, 6, and 10, and are sampled *after* training on all 10 concepts. See Appendix J for source of target images.

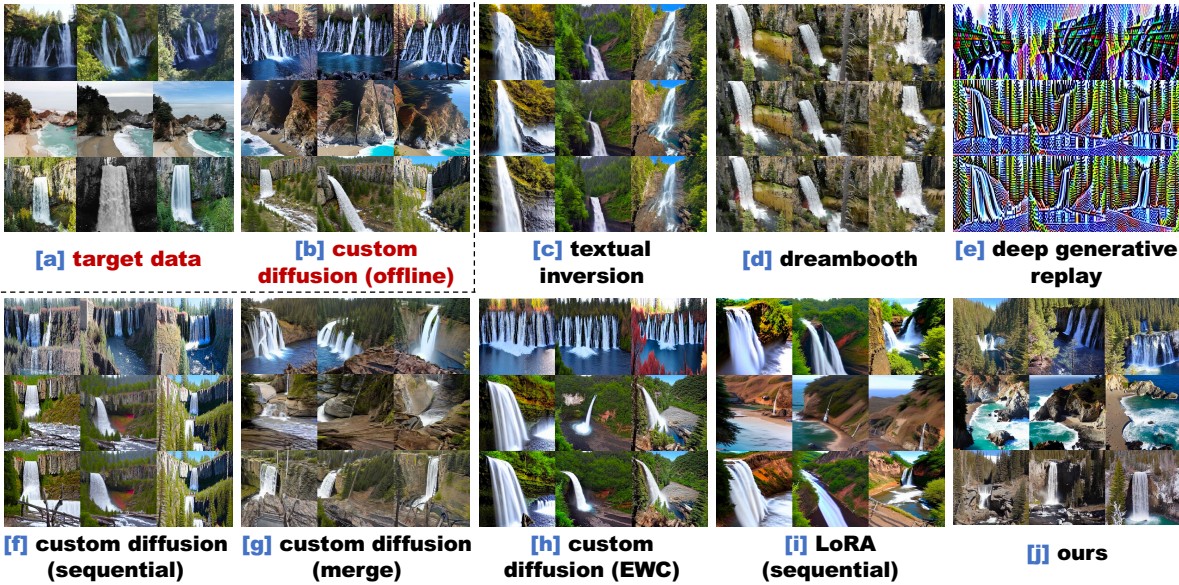

Figure 5: Qualitative results of continual customization using waterfalls from Google Landmarks (Weyand et al., 2020). Results are shown for concepts (tasks) 1, 5, and 10, and are sampled *after* training on all 10 concepts. See Appendix J for source of target images.

America *waterfall* landmarks which had at least 15 example images in the dataset. We chose waterfall landmarks given that they have *unique* characteristics yet are similar overall, making them a challenging task sequence (though not as challenging as human faces). Qualitative results are shown in Figure 5 with samples from task 1, 5, and 10 *after training all 10 tasks*, while quantitative results are given in Table 2. Overall, we notice similar trends to the face datasets, but custom diffusion (merge) performs better in this type of dataset compared to face datasets (though still much worse than our method). Specifically, it catastrophically forgets the middle landmark, but the first and final landmarks are represented well relative

Table 1: **Results on Celeb-A HQ (Karras et al., 2017; Liu et al., 2015).** $A_{mmd}$ ($\downarrow$) gives the average MMD score ($\times 10^3$) after training on all concept tasks, and $F_{mmd}$ ($\downarrow$) gives the average forgetting. Image-alignment give the CLIP-score between generated images after training on all tasks and dataset images. $N_{param}$ ($\downarrow$) gives the % number of parameters being trained and stored. CD represents Custom Diffusion (Kumari et al., 2022). We report the mean and standard deviation over 3 trials.

| Method | $N_{param}$ ($\downarrow$) Train | $N_{param}$ ($\downarrow$) Store | $A_{mmd}$ ($\downarrow$) | $F_{mmd}$ ($\downarrow$) | Image ($\uparrow$) alignment |
|---|---|---|---|---|---|
| Textual Inversion | 0.0 | 100.0 | $2.98 \pm 0.60$ | $\mathbf{0.00 \pm 0.00}$ | $0.69 \pm 0.02$ |
| DreamBooth | 100.0 | 100.0 | $9.01 \pm 0.73$ | $8.25 \pm 1.44$ | $0.54 \pm 0.02$ |
| Gen. Replay | 2.23 | 100.0 | $17.34 \pm 0.33$ | $11.91 \pm 0.31$ | $0.47 \pm 0.01$ |
| LoRA (Sequential) | 0.09 | 100.0 | $5.57 \pm 0.29$ | $3.96 \pm .112$ | $0.69 \pm .01$ |
| CD (Sequential) | 2.23 | 100.0 | $7.72 \pm .26$ | $7.08 \pm 0.54$ | $0.63 \pm 0.01$ |
| CD (Merge) | 2.23 | 122.30 | $8.89 \pm 1.59$ | $6.78 \pm 1.11$ | $0.53 \pm 0.04$ |
| CD EWC | 2.23 | 102.23 | $4.64 \pm 0.09$ | $2.95 \pm 0.07$ | $0.73 \pm 0.00$ |
| ZipLoRA | 0.10 | 101.00 | $3.25 \pm 0.04$ | $0.27 \pm 0.05$ | $0.76 \pm 0.01$ |
| Ours | 0.09 | 100.90 | $\mathbf{2.04 \pm 0.17}$ | $0.32 \pm 0.13$ | $\mathbf{0.80 \pm 0.01}$ |

Table 2: **Results on Google Landmarks dataset v2 (Weyand et al., 2020).** $A_{mmd}$ ($\downarrow$) gives the average MMD score ($\times 10^3$) after training on all concept tasks, and $F_{mmd}$ ($\downarrow$) gives the average forgetting. Image-alignment give the CLIP-score between generated images after training on all tasks and dataset images. $N_{param}$ ($\downarrow$) gives the % number of parameters being trained and stored. CD represents Custom Diffusion (Kumari et al., 2022). We report the mean and standard deviation over 3 trials.

| Method | $N_{param}$ ($\downarrow$) Train | $N_{param}$ ($\downarrow$) Store | $A_{mmd}$ ($\downarrow$) | $F_{mmd}$ ($\downarrow$) | Image ($\uparrow$) alignment |
|---|---|---|---|---|---|
| Textual Inversion | 0.0 | 100.0 | $3.84 \pm 0.55$ | $\mathbf{0.00 \pm 0.00}$ | $0.81 \pm 0.01$ |
| DreamBooth | 100.0 | 100.0 | $5.49 \pm 0.69$ | $4.70 \pm 0.39$ | $0.77 \pm 0.00$ |
| Gen. Replay | 2.23 | 100.0 | $14.83 \pm 0.85$ | $10.54 \pm 0.59$ | $0.55 \pm 0.00$ |
| LoRA (Sequential) | 0.09 | 100.0 | $4.44 \pm 0.48$ | $3.37 \pm 0.69$ | $0.79 \pm 0.00$ |
| CD (Sequential) | 2.23 | 100.0 | $5.04 \pm 0.74$ | $4.19 \pm 0.75$ | $0.78 \pm 0.02$ |
| CD (Merge) | 2.23 | 122.30 | $7.59 \pm 3.21$ | $5.14 \pm 1.79$ | $0.70 \pm 0.07$ |
| CD EWC | 2.23 | 102.23 | $4.61 \pm 0.41$ | $3.25 \pm 0.81$ | $0.79 \pm 0.01$ |
| ZipLoRA | 0.10 | 101.00 | $3.24 \pm 0.22$ | $0.51 \pm 0.21$ | $0.82 \pm 0.01$ |
| Ours | 0.09 | 100.90 | $\mathbf{2.82 \pm 0.26}$ | $0.30 \pm 0.27$ | $\mathbf{0.83 \pm 0.01}$ |

to the face datasets. **We emphasize that, in addition to being the top-performer, our method has significant parameter storage advantages over custom diffusion (merge).** Specifically, in order to create "on the fly" merged models, custom diffusion (merge) requires that KV values of all tasks be stored indefinitely, and has storage costs that are $25\times$ greater than our method for this 10 task sequence. However, we note that merging in in a "continual manner" would not incur this large storage costs. We discuss this more in Appendix C.

### 4.3 Multi-Concept Generations

In Figure 6, we provide some results demonstrating the ability to generate photos of multiple concepts in the same picture. We found that using the prompt style "a photo of V* person. Posing with V* person" worked best. We notice that only our *full* method has success on each concept pair. The two ablations (*ablate token init* removes the random token initialization and *ablate prompt concept* uses the prompt "a photo of V* person" instead of "a photo of $V^*$") are "hit or miss", whereas custom diffusion (merge) completely fails (we found the sequential version also fails due to catastrophic forgetting). One drawback is that we noticed difficulty in producing multi-concept images of individuals of highest similarity, which needs to be addressed in future work.

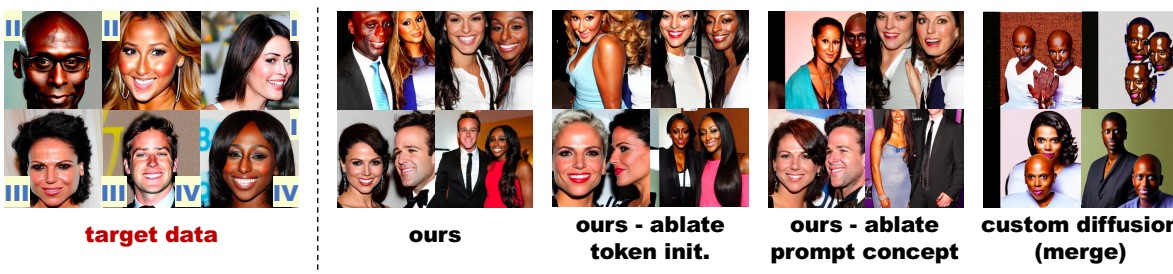

target data          ours          ours - ablate          ours - ablate          custom diffusion
                                    token init.            prompt concept         (merge)

Figure 6: Multi-concept results *after training on 10 sequential tasks* using Celeb-A HQ (Karras et al., 2017; Liu et al., 2015). Using standard quadrant numbering (I is upper right, II is upper left, III is lower left, IV is lower right), we label which target data belongs in which generated image by directly annotating the target data images. See Appendix J for source of target images.

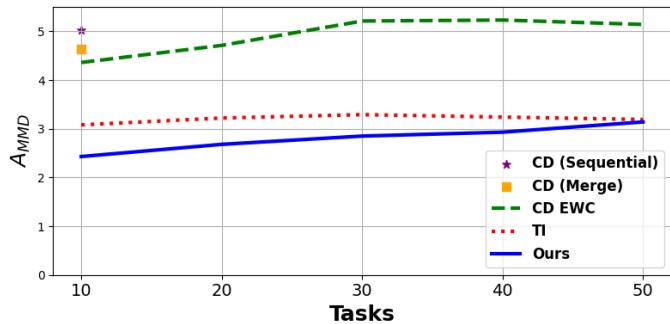

Figure 7: Analysis on extending C-LoRA to a 50 concept sequence comprised of both Celeb-A HQ (Karras et al., 2017; Liu et al., 2015) and Google Landmarks dataset v2 (Weyand et al., 2020) shows that our method struggles with long concept sequences, eventually performing worse than Textual Inversion (Gal et al., 2022).

## 5    On Longer Task Sequences

In Figure 7, we explore the scalability of C-LoRA by extending it to handle a sequence of 50 concepts, drawing from both the Celeb-A HQ dataset (Karras et al., 2017; Liu et al., 2015) and the Google Landmarks dataset v2 (Weyand et al., 2020). We found that, while C-LoRA shows promise in handling a moderate number of concepts, its performance begins to falter as the concept sequence lengthens. Specifically, it approaches the performance of Textual Inversion (Gal et al., 2022), a very simple method which does not alter the backbone of the text-to-image model (and instead only learns custom tokens). This suggests there is a need for further refinement to enhance scalability and effectiveness across broader and more complex concept sequences for large-scale continual learning tasks. For further context, we include the results of Custom Diffusion[4] (Kumari et al., 2022) with EWC (Kirkpatrick et al., 2017) regularization, showing that our method still outperforms this approach. **We include additional results in our Appendix**, including additional metrics and results (Appendix A), additional method ablations (Appendix B), empirical analysis on sequential update parameter interference (Appendix D), further analysis on the merge variant of Custom Diffusion (Kumari et al., 2022) (Appendix E), additional multi-concept generations (Appendix F), an offline comparison with Custom Diffusion (Kumari et al., 2022) (Appendix G), C-LoRA hyperparameter sweeps (Appendix H).

## 6    Continual Image Classification Experiments

The previous section shows that our C-LoRA achieves SOTA performance in the *proposed* setting of *continual customization of text-to-image diffusion*. While we compared with SOTA existing methods, and furthermore tried our best to engineer improved baselines, we want to strengthen our claims that our

---

[4]We do not include the full plots for unregularized Custom Diffusion, as they underperform Custom Diffusion with regularization.

Table 3: **Results (%) on ImageNet-R** for 10 tasks (20 classes per task). $A_N$ gives the accuracy averaged over tasks and $F_N$ gives the average forgetting. We report the mean and standard deviation over 3 trials.

| Method | $A_N$ ($\uparrow$) | $F_N$ ($\downarrow$) |
|---|---|---|
| UB (tune all) | 77.13 | - |
| UB (tune QKV) | 82.30 | - |
| FT | $10.12 \pm 0.51$ | $25.69 \pm 0.23$ |
| FT++ | $48.93 \pm 1.15$ | $9.81 \pm 0.31$ |
| LwF.MC | $66.73 \pm 1.25$ | $3.52 \pm 0.39$ |
| L2P | $69.00 \pm 0.72$ | $2.09 \pm 0.22$ |
| L2P++ | $70.98 \pm 0.47$ | $1.79 \pm 0.19$ |
| DualPrompt | $74.12 \pm 0.55$ | $1.60 \pm 0.20$ |
| CODA-P | $75.58 \pm 0.61$ | $1.65 \pm 0.12$ |
| Ours - Ablate Eq. 3 | $72.92 \pm 0.52$ | $3.73 \pm 0.06$ |
| **Ours** | $\mathbf{78.16 \pm 0.43}$ | $\mathbf{1.30 \pm 0.10}$ |

C-LoRA is indeed a SOTA technique for continual learning. Therefore, in this section we also demonstrate that our method can achieve SOTA performance in the *well-established* setting of *rehearsal-free continual learning for image classification.*

**Implementation Details.** We benchmark our approach using ImageNet-R (Hendrycks et al., 2021; Wang et al., 2022b) which is composed of 200 object classes with a wide collection of image styles, including cartoon, graffiti, and hard examples from the original ImageNet dataset (Russakovsky et al., 2015). This benchmark is attractive because the distribution of training data has significant distance to the pre-training data (ImageNet), thus providing a fair and challenging problem setting. In addition to the original 10-task benchmark, we also provide results with a smaller number of large tasks (5-task) and a larger number of small tasks (20 task) in Appendix I.

We use the exact same experiment setting as the recent CODA-Prompt (Smith et al., 2022b) paper. We implement our method and all baselines in PyTorch(Paszke et al., 2019) using the ViT-B/16 backbone (Dosovitskiy et al., 2020) pre-trained on ImageNet-1K (Russakovsky et al., 2015). We compare to the following methods (the same rehearsal-free comparisons of CODA-Prompt): Learning without Forgetting (LwF) (Li & Hoiem, 2017), Learning to Prompt (L2P) (Wang et al., 2022c), a modified version of L2P (L2P++) (Smith et al., 2022b), and DualPrompt (Wang et al., 2022b). Additionally, we report the upper bound (UB) performance, which is trained offline (we provide two variants: one fine-tunes all parameters and the other only fine-tunes QKV projection matrices of self-attention blocks), and performance for a neural network trained only on classification loss using the new task training data (we refer to this as FT). We use the same classification head as L2P, DualPrompt, and CODA-Prompt, and additionally compare to a FT variant, FT++, which uses the same classifier as the prompting methods and suffers from less forgetting. For additional details, we refer the reader to original CODA-Prompt (Smith et al., 2022b) paper. For our method, we add C-LoRA to the QKV projection matrices of self-attention blocks throughout the ViT model.

**Metrics.** We evaluate methods using (1) average accuracy $A_N$, or the accuracy with respect to all past classes averaged over $N$ tasks, and (2) average forgetting (Chaudhry et al., 2019a; Lopez-Paz & Ranzato, 2017) $F_N$, or the drop in task performance averaged over $N$ tasks. We emphasize that $A_N$ is the more important metric and encompasses both method plasticity *and* forgetting, whereas $F_N$ simply provides additional context.

**Results.** In Table 3, we benchmark our C-LoRA against the popular and recent rehearsal-free continual learning methods. **We found that our method achieves a high state-of-the-art in a setting that it was not designed for.** Compared to the prompting methods L2P, DualPrompt, and the recent CODA-Prompt, our method has clear and significant improvements. We also ablate our forgetting loss and show that this simple yet *major* contribution is the main driving force behind our performance.

## 7    Conclusion

In this paper, we addressed the problem of catastrophic forgetting in continual customization of text-to-image models (Continual Diffusion). Our proposed method, C-LoRA, leverages the structure of cross attention layers and adapts them to new concepts in a low-rank manner while preserving the knowledge of past concepts through a self-regularization mechanism. We showed that our approach alleviates catastrophic forgetting in the Stable Diffusion model with both quantitative and qualitative analysis. Additionally, we extended our method to the well-established setting of continual learning for image classification, demonstrating that our gains translate to state-of-the-art performance in the standard benchmark as well. Our contributions provide a promising solution for the practical and critical problem of continually customizing text-to-image models while avoiding interference and forgetting prior seen concepts.

## 8    Limitations & Broader Impact Statement

Despite the success of our approach in generating task sequences of up to ten faces, we acknowledge key limitations and cautions that must be addressed. Specifically, we caution against training our method on large task sequences of 50 or 100 faces, as our approach suffers in longer concept sequences (as shown in Figure 7). To increase the scalability of our approach, future work should focus on exploring additional novel techniques. Additionally, all existing methods (including ours which performs best) still struggle to generate multi-conceptual faces on similar individuals, and we strongly urge future research to focus on improving this aspect of our approach.

In selecting the CelebFaces Attributes (Celeb-A) HQ (Karras et al., 2017; Liu et al., 2015) and Google Landmarks dataset v2 (Weyand et al., 2020) for our benchmarks[5], we aimed to demonstrate the versatility and effectiveness of our approach across varied domains. The choice of the Celeb-A dataset was motivated by the unique challenge that human faces present in terms of variability in fine-grained features. Similarly, the selection of the Google Landmarks dataset, specifically North American waterfall landmarks, was intended to showcase our method's applicability to natural scenes, which, despite their unique characteristics, maintain a degree of similarity that poses a different kind of challenge. To avoid bias in the selection of specific concepts to use for our experiments, we randomly selected valid concept identities for all datasets, where validity was determined by having at least 15 individual training images each. We believe that future research should prioritize the use of similar datasets or consenting participants to ensure ethical practices.

Furthermore, we emphasize the ethical implications of our work and caution against the use of our approach to generate faces of individuals who do not consent. We specifically acknowledge the potential for C-LoRA to contribute to the creation of realistic images that blend multiple concepts or identities. This capability raises critical concerns regarding the potential for misuse in creating disinformation or harming individuals' reputations through unauthorized or deceptive representations. We therefore stress the importance of developing and adhering to ethical guidelines and regulatory frameworks that specifically address these risks, ensuring that advancements, such as ours, are used responsibly and with a clear commitment to upholding individual privacy and integrity.

Finally, we include a strong disclaimer against using our work in production environments without stringent ethical oversight and within the bounds of legal frameworks designed to protect individual privacy and integrity. Furthermore, we acknowledge the ethics concerns over using artists' images. We avoid artistic creativity in our work, and note that this is being worked out in the policy and legal worlds. Despite these cautions, we believe that our work has the potential to bring great joy to individuals in the form of mobile app entertainment. By scaling the generation of faces to multiple sequential faces, our approach enables novel entertainment applications for sequential settings. We are confident that our approach will pave the way for exciting advancements in the field of text-to-image generation.

---

[5]We refer the reader to Appendix J for more information on the image sources for figures.

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

# Appendix

## A Additional Metrics and Results

Tables A and B extend the main paper results with additional metrics and results. In the original paper, we report: (i) $N_{param}$, the number of parameters *trained* (i.e., unlocked during training a task) in % of the U-Net backbone model, (ii) $A_{mmd}$, the average MMD score ($\times 10^3$) after training on all concept tasks, and (iii) $F_{mmd}$, average forgetting. Consider $N$ customization "tasks" $t \in \{1, 2, \ldots, N-1, N\}$, where $N$ is the total number of concepts that will be shown to our model. We denote $X_{i,j}$ as task $j$ images generated by the model after training task $i$. Furthermore, we denote $X_{D,j}$ as original dataset images for task $j$. Using these terms, we calculate our $A_{mmd}$ metric (where lower is better) as:

$$A_{mmd} = \frac{1}{N} \sum_{j=1}^{N} MMD\left(\mathcal{F}_{clip}(X_{D,j}), \mathcal{F}_{clip}(X_{N,j})\right) \tag{4}$$

where $\mathcal{F}_{clip}$ denotes a function embedding into a rich semantic space using CLIP (Radford et al., 2021), and $MMD$ denotes the Maximum Mean Discrepancy (Gretton et al., 2012) with a polynomial kernel. To calculate our forgetting metric $F_{mmd}$, we calculate the average distance the images have changed over training, or:

$$F_{mmd} = \frac{1}{N-1} \sum_{j=1}^{N-1} MMD\left(\mathcal{F}_{clip}(X_{j,j}), \mathcal{F}_{clip}(X_{N,j})\right) \tag{5}$$

In addition to these metrics, we include metrics reported from Kumari et al. (2022): (iv) *image-alignment*, which calculates the image-image CLIP-Score (Gal et al., 2022) between all $X_{D,j}$ and $X_{N,j}$, (v) *text-alignment*, which calculates the image-text CLIP-Score (Hessel et al., 2021) between all $X_{N,j}$ and the prompt "a *object*", and finally (vi) KID (Bińkowski et al., 2018) score ($\times 1e3$). We notice that the metrics mostly tell a similar and intuitive story to our analysis from the main text, with one main difference: methods with high catastrophic forgetting seem to have good or better *text-alignment* than our method - this can be attributed to this metric having no conditioning on the *individual concepts*, just the general object (e.g., a "face" or "waterfall"). We include this metric for consistency for the related setting of *offline* (i.e., not continual) customization of text-to-image models, but we do not believe it is insightful for our setting of Continual Diffusion.

## B Ablation Studies

We include an ablation study using the CelebFaces Attributes (Celeb-A) HQ (Karras et al., 2017; Liu et al., 2015) benchmark in Table C. We ablate 3 key components of our method: (i) *ablate token init* removes the random token initialization, (ii) *ablate prompt concept* uses the prompt "a photo of $V^*$ person" instead of "a photo of $V^*$", and (iii) *ablate Eq. 3* removes the forgetting loss from our method. (ii) and (iii) have clear performance drops compared to the full method. At first glance, (i) (removing the random token initialization) seems to actually help performance, *but, as we show in Section 4.3 (Figure 6), this design component is crucial for **multi-concept generations**, and without it our method struggles to correctly generate multi-concept image samples.*

Table A: **Full results on Celeb-A HQ (Karras et al., 2017; Liu et al., 2015)** after training 10 concepts sequentially. $N_{param}$ ($\downarrow$) gives the % number of parameters being trained and stored, $A_{mmd}$ ($\downarrow$) gives the average MMD score ($\times 10^3$) after training on all concept tasks, $F_{mmd}$ ($\downarrow$) gives the average forgetting, Image/Text-alignment give the CLIP-score between generated images after training on all tasks and dataset images/captions, respectively, and KID gives the Kernel Inception Distance ($\times 10^3$) between generated and dataset images. CD represents Custom Diffusion (Kumari et al., 2022). We report the mean and standard deviation over 3 trials.

| Method | $N_{param}$ ($\downarrow$) Train | $N_{param}$ ($\downarrow$) Store | $A_{mmd}$ ($\downarrow$) | $F_{mmd}$ ($\downarrow$) | Image ($\uparrow$) alignment | Text ($\uparrow$) alignment | KID ($\downarrow$) |
|---|---|---|---|---|---|---|---|
| Textual Inversion | 0.0 | 100.0 | $2.98 \pm 0.60$ | $\mathbf{0.00 \pm 0.00}$ | $0.69 \pm 0.02$ | $0.53 \pm 0.01$ | $49.69 \pm 14.45$ |
| DreamBooth | 100.0 | 100.0 | $9.01 \pm 0.73$ | $8.25 \pm 1.44$ | $0.54 \pm 0.02$ | $\mathbf{0.60 \pm 0.01}$ | $173.48 \pm 34.03$ |
| Gen. Replay | 2.23 | 100.0 | $17.34 \pm 0.33$ | $11.91 \pm 0.31$ | $0.47 \pm 0.01$ | $\mathbf{0.60 \pm 0.01}$ | $422.11 \pm 12.19$ |
| LoRA (Sequential) | 0.09 | 100.0 | $5.57 \pm 0.29$ | $3.96 \pm .112$ | $0.69 \pm .01$ | $0.56 \pm 0.01$ | $81.67 \pm 12.11$ |
| CD (Sequential) | 2.23 | 100.0 | $7.72 \pm .26$ | $7.08 \pm 0.54$ | $0.63 \pm 0.01$ | $0.58 \pm 0.01$ | $90.48 \pm 7.56$ |
| CD (Merge) | 2.23 | 122.30 | $8.89 \pm 1.59$ | $6.78 \pm 1.11$ | $0.53 \pm 0.04$ | $0.56 \pm 0.01$ | $139.27 \pm 43.78$ |
| CD EWC | 2.23 | 102.23 | $4.64 \pm 0.09$ | $2.95 \pm 0.07$ | $0.73 \pm 0.00$ | $0.56 \pm 0.01$ | $39.09 \pm 2.15$ |
| ZipLoRA | 0.10 | 101.00 | $3.25 \pm 0.04$ | $0.27 \pm 0.05$ | $0.76 \pm 0.01$ | $0.55 \pm 0.00$ | $40.30 \pm 7.20$ |
| Ours | 0.09 | 100.90 | $\mathbf{2.04 \pm 0.17}$ | $0.32 \pm 0.13$ | $\mathbf{0.80 \pm 0.01}$ | $0.55 \pm 0.00$ | $\mathbf{26.50 \pm 1.53}$ |

Table B: **Full results on Google Landmarks dataset v2 (Weyand et al., 2020)** after training 10 concepts sequentially. $N_{param}$ ($\downarrow$) gives the % number of parameters being trained and stored, $A_{mmd}$ ($\downarrow$) gives the average MMD score ($\times 10^3$) after training on all concept tasks, $F_{mmd}$ ($\downarrow$) gives the average forgetting, Image/Text-alignment give the CLIP-score between generated images after training on all tasks and dataset images/captions, respectively, and KID gives the Kernel Inception Distance ($\times 10^3$) between generated and dataset images. CD represents Custom Diffusion (Kumari et al., 2022). We report the mean and standard deviation over 3 trials.

| Method | $N_{param}$ ($\downarrow$) Train | $N_{param}$ ($\downarrow$) Store | $A_{mmd}$ ($\downarrow$) | $F_{mmd}$ ($\downarrow$) | Image ($\uparrow$) alignment | Text ($\uparrow$) alignment | KID ($\downarrow$) |
|---|---|---|---|---|---|---|---|
| Textual Inversion | 0.0 | 100.0 | $3.84 \pm 0.55$ | $\mathbf{0.00 \pm 0.00}$ | $0.81 \pm 0.01$ | $0.64 \pm 0.02$ | $46.28 \pm 8.99$ |
| DreamBooth | 100.0 | 100.0 | $5.49 \pm 0.69$ | $4.70 \pm 0.39$ | $0.77 \pm 0.00$ | $0.68 \pm 0.01$ | $110.13 \pm 15.79$ |
| Gen. Replay | 2.23 | 100.0 | $14.83 \pm 0.85$ | $10.54 \pm 0.59$ | $0.55 \pm 0.00$ | $0.65 \pm 0.00$ | $491.88 \pm 8.32$ |
| LoRA (Sequential) | 0.09 | 100.0 | $4.44 \pm 0.48$ | $3.37 \pm 0.69$ | $0.79 \pm 0.00$ | $0.67 \pm 0.00$ | $89.18 \pm 9.36$ |
| CD (Sequential) | 2.23 | 100.0 | $5.04 \pm 0.74$ | $4.19 \pm 0.75$ | $0.78 \pm 0.02$ | $\mathbf{0.69 \pm 0.02}$ | $141.45 \pm 77.12$ |
| CD (Merge) | 2.23 | 122.30 | $7.59 \pm 3.21$ | $5.14 \pm 1.79$ | $0.70 \pm 0.07$ | $0.61 \pm 0.05$ | $141.45 \pm 77.12$ |
| CD EWC | 2.23 | 102.23 | $4.61 \pm 0.41$ | $3.25 \pm 0.81$ | $0.79 \pm 0.01$ | $\mathbf{0.69 \pm 0.02}$ | $95.54 \pm 12.23$ |
| ZipLoRA | 0.10 | 101.00 | $3.24 \pm 0.22$ | $0.51 \pm 0.21$ | $0.82 \pm 0.01$ | $0.65 \pm 0.01$ | $71.25 \pm 8.01$ |
| Ours | 0.09 | 100.90 | $\mathbf{2.82 \pm 0.26}$ | $0.30 \pm 0.27$ | $\mathbf{0.83 \pm 0.01}$ | $0.58 \pm 0.03$ | $\mathbf{34.43 \pm 1.81}$ |

Table C: **Ablation results on Celeb-A HQ (Karras et al., 2017; Liu et al., 2015)** after training 10 concepts sequentially. $N_{param}$ ($\downarrow$) gives the % number of parameters being trained and stored, $A_{mmd}$ ($\downarrow$) gives the average MMD score ($\times 10^3$) after training on all concept tasks, $F_{mmd}$ ($\downarrow$) gives the average forgetting, Image/Text-alignment give the CLIP-score between generated images after training on all tasks and dataset images/captions, respectively, and KID gives the Kernel Inception Distance ($\times 10^3$) between generated and dataset images. We report the mean and standard deviation over 3 trials.

| Method | $N_{param}$ ($\downarrow$) Train | $N_{param}$ ($\downarrow$) Store | $A_{mmd}$ ($\downarrow$) | $F_{mmd}$ ($\downarrow$) | Image ($\uparrow$) alignment | Text ($\uparrow$) alignment | KID ($\downarrow$) |
|---|---|---|---|---|---|---|---|
| Ours | 0.09 | 100.90 | $2.04 \pm 0.17$ | $0.32 \pm 0.13$ | $\mathbf{0.80 \pm 0.01}$ | $0.55 \pm 0.00$ | $26.50 \pm 1.53$ |
| (i) Ablate token init | 0.09 | 100.90 | $\mathbf{1.95 \pm 0.18}$ | $\mathbf{0.20 \pm 0.13}$ | $0.79 \pm 0.01$ | $0.54 \pm 0.00$ | $34.96 \pm 0.14$ |
| (ii) Ablate prompt concept | 0.09 | 100.90 | $2.41 \pm 0.42$ | $0.54 \pm 0.39$ | $0.78 \pm 0.02$ | $\mathbf{0.53 \pm 0.01}$ | $\mathbf{18.95 \pm 0.25}$ |
| (i) + (ii) | 0.09 | 100.90 | $2.76 \pm 0.26$ | $0.23 \pm 0.15$ | $0.78 \pm 0.11$ | $0.54 \pm 0.00$ | $23.96 \pm 0.51$ |
| (iii) Ablate Eq. 3 | 0.09 | 100.90 | $5.12 \pm 0.06$ | $4.15 \pm 0.13$ | $0.69 \pm 0.09$ | $0.56 \pm 0.01$ | $90.16 \pm 1.96$ |

## C  Additional Implementation Details

We use 2 A100 GPUs to generate all results. All hyperparameters mentioned were searched with an exponential search (for example, learning rates were chosen in the range $5e-2, 5e-3, 5e-4, 5e-5, 5e-6, 5e-7, 5e-8$; we directly show the $\lambda$ and rank sweeps in Figure H). We did not tune hyperparameters on the full task set because tuning hyperparameters with hold out data from all tasks may violate the principal of continual learning that states each task is visited only once (van de Ven & Tolias, 2019); instead, we used a different task sequence of size 5. We report the mean and standard deviation over 3 trials containing different randomly sampled task data. We found a learning rate of $5e-6$ worked best for all non-LoRA methods, and a learning rate of $5e-4$ worked best for our LoRA methods. We found a loss weight of $1e6$ and $1e8$ worked best for EWC (Kirkpatrick et al., 2017) and C-LoRA respectively. These values are high because the importance weighting factors are much smaller than 1, and squared, and thus the loss penalty is a very small number and is not effective without a large loss weighting. We found a rank of 16 was sufficient for LoRA for the text-to-image experiments, and 64 for the image classification experiments. These were chosen from a range of $8, 16, 32, 64, 128$. All images are generated with a 512x512 resolution, and we train for 2000 steps on the face datasets and 500 steps on the waterfall datasets. For additional analysis on the efficacy for LoRA (Hu et al., 2021) for text-to-image diffusion, we suggest the implementation by Ryu, which was a concurrent project to ours.

**On stored parameters in our implementation:** For our method, the number of parameters trained is 0.09% of the U-Net backbone. In contrast, this number is 2.23% for Custom Diffusion (Kumari et al., 2022). For storage, we have two options for our method: store the individual LoRA weights for each past concept or store the summation. For a 10-concept sequence, the storage cost would be 0.09% * 10 = 0.9%. However, *we note that our storage is bounded by the size of the summed multiplied matrices.* Thus, if the sum of individual LoRA weights exceeds the size of the summed multiplied matrix, we only store the summed multiplied matrix long term, and thus our max cost is equivalent to the 2.23% of Custom Diffusion.

We note that there are two ways to implement Custom Diffusion (Merge) (Kumari et al., 2022) for continual learning. One way is to store the individual weights for each concept and then merge them together at inference time (as done in the original paper). A second way would be to merge the weights online without storing the individual concept weights. We chose the first option as it performs better and is a more faithful implementation of their method. As a result, the the parameters stored is much higher for Custom Diffusion (Merge) than all other methods, including ours.

## D  A Closer Look at LoRA and Interference

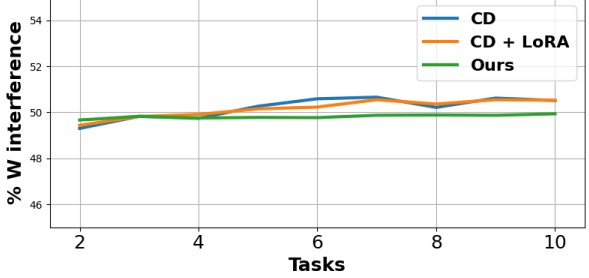
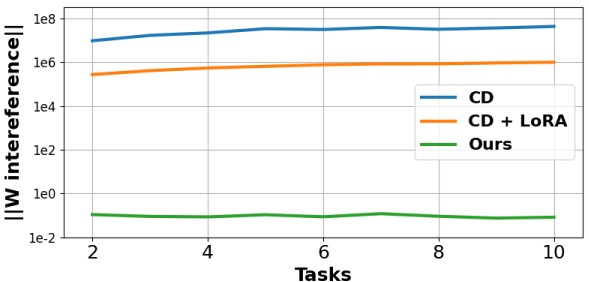

(a) *Percent* of weights that are updated in the opposite direction as the prior task (i.e., *interference*)

(b) *Amount* of change in the opposite direction as the prior task (i.e., the *magnitude* of interference)

Figure A: **Sequential interference analysis** (Table 1 setting).

In Figure A, we analyze sequential update interference for CD (Kumari et al., 2022), CD+LoRA, and ours. The left plot shows the % of weights that are updated in the opposite direction as the prior task (i.e., *interference*), and we see our method has the least directional interference; however, the major difference lies in the right plot, which shows the *amount* of change in the opposite direction as the prior task (i.e., the

*magnitude* of interference), and we see 1) LoRA reduces this magnitude, motivating its use in CL, and 2) our method reduces this magnitude much further.

# E  A Closer Look at Custom Diffusion Merge

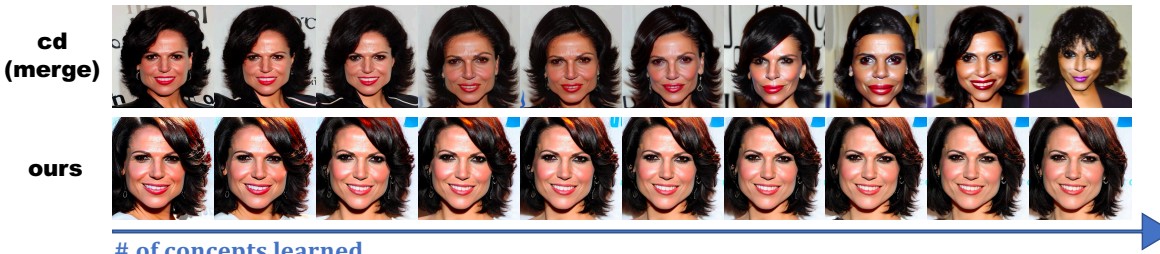

Figure B: **Analysis of Custom Diffusion (CD) - Merge (Kumari et al., 2022) vs ours.** We show generations of concept 1, starting after concept 1 is learned (far left) and ending with after concept 10 is learned (far right). These results show that CD (Merge) forgets concept 1 over time, whereas our method does not.

In Figure B, we analyze Custom Diffusion (CD) - Merge (Kumari et al., 2022) vs our method through learning the ten concept sequence in Celeb-A HQ (Karras et al., 2017; Liu et al., 2015). On the far left, we see a generated example of concept 1 after learning concept 1. On the far right, we see a generated example of concept 1 after learning concept 10. We see that, for CD (Merge), the identity is retained through the first 6 concepts, and then concepts begin to interfere after 7 concepts have been learned. On the other hand, our method generates good samples of concept 1 for the entire sequence. The takeaway from this analysis is that CD (Merge) is only effective for a hand-full of concepts, unlike our method. This trend extends to other datasets - CD (Merge) typically performs similarly to our method after 5 concepts, but has severe performance decreases before the 10th concept.

# F  Additional Multi-Concept Generations

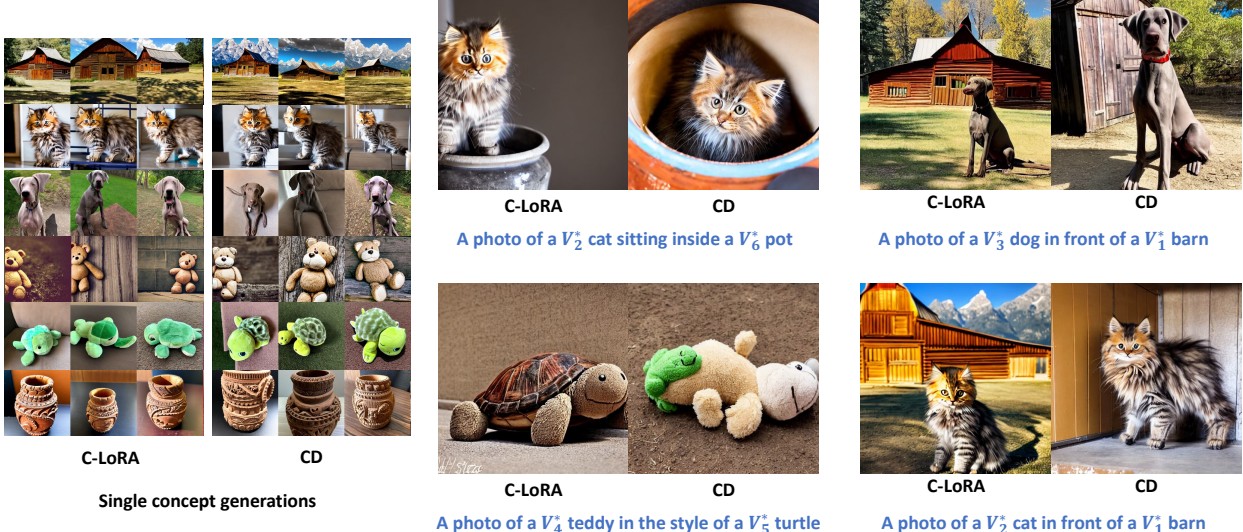

Figure C: Multi-concept results *after training on 6 sequential tasks* using CustomConcept101 dataset (Kumari et al., 2022). CD represents Custom Diffusion (Kumari et al., 2022).

In Figure C, we present additional results showcasing the capability of generating images incorporating multiple concepts simultaneously, utilizing six random concepts from the simple CustomConcept101 dataset (Kumari et al., 2022). **We wish to clarify that this dataset is not the primary focus of our method, as it lacks closely related, fine-grained concepts.** These results are offered to illustrate that our method achieves competitive performance in scenarios for which it was not specifically tailored. It is observed that both Custom Diffusion (Merge) (Kumari et al., 2022) and our C-LoRA can produce reasonable images featuring multiple concepts, following training across all six sequential tasks. However, we do note that the identity of the second concept in the prompt tends to be lost in the continual learning scenario for CD (Merge), despite verifying that single-concept generations from both methods perform similarly to the results presented in Kumari et al. (2022).

## G   Offline Comparison with Custom Diffusion

Table D: **Offline results using a combination of Celeb-A HQ (Karras et al., 2017; Liu et al., 2015) and Google Landmarks dataset v2 (Weyand et al., 2020)**. $A_{mmd}$ ($\downarrow$) gives the average MMD score ($\times 10^3$) after training on all concept tasks, Image/Text-alignment give the CLIP-score between generated images after training on all tasks and dataset images/captions, respectively, and KID gives the Kernel Inception Distance ($\times 10^3$) between generated and dataset images. CD represents Custom Diffusion (Kumari et al., 2022). We report the mean only across 6 concepts.

| Method | $A_{mmd}$ ($\downarrow$) | Image ($\uparrow$) alignment | Text ($\uparrow$) alignment | KID ($\downarrow$) |
|--------|--------------------------|------------------------------|-----------------------------|--------------------|
| Ours   | 2.15                     | 0.83                         | 0.59                        | **34.81**          |
| CD     | **2.12**                 | **0.84**                     | **0.61**                    | 39.37              |

In Table D, we compare the "starting place" of our method vs. Custom Diffusion (CD) (Kumari et al., 2022) to ensure a fair comparison. Specifically, we report the relevant offline learning metrics for the first concept of all 6 combined trials in Tables A/B. We see the CD actually performs better in this simple experiment, demonstrating that our gains are not due simply to an unfair advantage such as using LoRA (Hu et al., 2021).

## H   A Closer Look at Hyperparameters

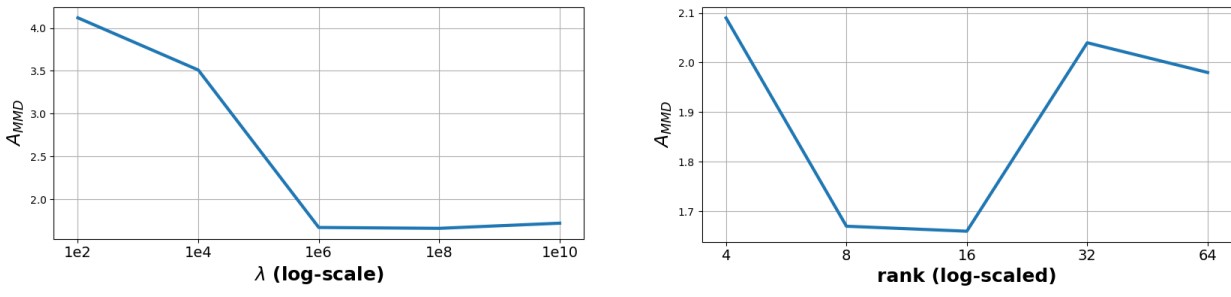

Figure D: **Hyperparameter analysis** (Table 1 setting, using only the first 5 concepts).

In Figure D, we delve into the impact of varying the $\lambda$ and rank values on the performance of our C-LoRA method. Our analysis reveals a range of effective values for both hyperparameters, bounded by regions where performance noticeably declines.

Table E: **Results (%) on ImageNet-R** for 5 tasks (40 classes per task), 10 tasks (20 classes per task), and 20 tasks (10 classes per task). $A_N$ gives the accuracy averaged over tasks and $F_N$ gives the average forgetting. We report the mean and standard deviation over 3 trials.

| Tasks | 5 | | 10 | | 20 | |
|---|---|---|---|---|---|---|
| Method | $A_N$ ($\uparrow$) | $F_N$ ($\downarrow$) | $A_N$ ($\uparrow$) | $F_N$ ($\downarrow$) | $A_N$ ($\uparrow$) | $F_N$ ($\downarrow$) |
| UB (tune all) | 77.13 | - | 77.13 | - | 77.13 | - |
| UB (tune QKV) | 82.30 | - | 82.30 | - | 82.30 | - |
| FT | $18.74 \pm 0.44$ | $41.49 \pm 0.52$ | $10.12 \pm 0.51$ | $25.69 \pm 0.23$ | $4.75 \pm 0.40$ | $16.34 \pm 0.19$ |
| FT++ | $60.42 \pm 0.87$ | $14.66 \pm 0.24$ | $48.93 \pm 1.15$ | $9.81 \pm 0.31$ | $35.98 \pm 1.38$ | $6.63 \pm 0.11$ |
| LwF.MC | $74.56 \pm 0.59$ | $4.98 \pm 0.37$ | $66.73 \pm 1.25$ | $3.52 \pm 0.39$ | $54.05 \pm 2.66$ | $2.86 \pm 0.26$ |
| L2P | $70.25 \pm 0.40$ | $3.15 \pm 0.37$ | $69.00 \pm 0.72$ | $2.09 \pm 0.22$ | $65.70 \pm 1.35$ | $1.27 \pm 0.18$ |
| L2P++ | $71.95 \pm 0.38$ | $2.52 \pm 0.38$ | $70.98 \pm 0.47$ | $1.79 \pm 0.19$ | $67.24 \pm 1.05$ | $1.12 \pm 0.21$ |
| DualPrompt | $72.85 \pm 0.39$ | $2.49 \pm 0.36$ | $71.49 \pm 0.66$ | $1.67 \pm 0.29$ | $68.02 \pm 1.38$ | $1.07 \pm 0.18$ |
| CODA-P (small) | $75.38 \pm 0.20$ | $2.65 \pm 0.15$ | $74.12 \pm 0.55$ | $1.60 \pm 0.20$ | $70.43 \pm 1.30$ | $1.00 \pm 0.15$ |
| CODA-P | $76.41 \pm 0.38$ | $2.98 \pm 0.20$ | $75.58 \pm 0.61$ | $1.65 \pm 0.12$ | $\mathbf{72.08 \pm 1.03}$ | $\mathbf{0.97 \pm 0.16}$ |
| Ours - Ablate Eq. 3 | $76.62 \pm 0.16$ | $5.21 \pm 0.19$ | $72.92 \pm 0.52$ | $3.73 \pm 0.06$ | $65.08 \pm 0.88$ | $4.93 \pm 0.11$ |
| **Ours** | $\mathbf{79.30 \pm 0.34}$ | $\mathbf{2.45 \pm 0.30}$ | $\mathbf{78.16 \pm 0.43}$ | $\mathbf{1.30 \pm 0.10}$ | $71.85 \pm 0.13$ | $1.65 \pm 0.39$ |

## I Additional Results for ImageNet-R

In Table E, we extend the image classification results to include smaller (5) and longer (20) task sequences. We see that, for the most part, all trends hold. However, we notice that our C-LoRA method slightly under-performs CODA-Prompt (Smith et al., 2022b) for the long task sequence (20). We note that prompting methods such as CODA-Prompt grow in number of total parameters with each additional task, whereas our model is constant sized (the small number LoRA parameters collapse back into the model during inference). It is possible that our method would perform better on long task sequences with a larger LoRA rank.

## J On Image Sources for the Figures

Due to licensing constraints for sharing dataset images, we instead generated similar images to display in many of our figures. In Figures 1, 4, and 6, we generated the images labeled "target data" with single-concept Custom Diffusion (Kumari et al., 2022) using LoRA (Hu et al., 2021) adaptation (we refer to these as *pseudo figure images* for the rest of this section). **We emphasize that all training and evaluation have been performed using the original datasets, and the result images were obtained through models trained directly on these datasets**. For Figure 2, the images captioned "learn" are *pseudo figure images*, and the multi-concept images are results produced with our method. For Figure 3, all face images are *pseudo figure images*. In Figure 5, the "target data" were collected from `https://openverse.org/` and are all licensed with either a CC-0 license or marked for public domain. Figures B/ C only contain results produced from models we trained.

