# OpenReview forum: "Continual Diffusion: Continual Customization of Text-to-Image Diffusion with C-LoRA"
_TMLR — Accepted by TMLR_

### Review · Reviewer_VgvP · 2024-01-10

**Summary Of Contributions:**

This paper studies the continual setup of concept learning from text-to-image diffusion models. The proposed method, C-LoRA, learns a LoRA for each concept, which is regularized to prevent catastrophic forgetting. C-LoRA outperforms the previous multi-concept learning approach like Custom Diffusion, both qualitatively and quantitatively.

**Audience:**

Yes

**Broader Impact Concerns:**

Well discussed.

**Claims And Evidence:**

Yes

**Requested Changes:**

Despite the limitations, I believe the paper is worth presenting in TMLR. Nevertheless, including a comparison with ZipLoRA would be informative, as the experimental cost would not be that expensive.

**Strengths And Weaknesses:**

## Strength

- C-LoRA is intuitively designed for continual multi-concept learning.
- C-LoRA outperforms the previous multi-concept learning method, Custom Diffusion.


## Weakness

- Continual learning of multiple concepts is not very practical.
	- In the most real scenario, people do not learn the concepts sequentially. Instead, different individuals create their own LoRAs, which are combined by the user in a post-hoc manner. Thus, Custom Diffusion (merge) and ZipLoRA (ZipLoRA: Any Subject in Any Style by Effectively Merging LoRAs) would be more practical.
- Limited technical novelty.
	- C-LoRA learns a LoRA for each concept, which is a common practice. The performance gain comes from preventing catastrophic forgetting, as this paper considers a new setup for continual learning. Thus, it is not surprising that C-LoRA outperforms competitors that are not designed for this continual setup.

---

### Review · Reviewer_Nanp · 2024-01-17

**Summary Of Contributions:**

This paper proposes a method to allow text-to-image generation models (e.g., Stable Diffusion) to learn multiple concepts (provided as images) in a sequential manner. They do this effectively by introducing C-LoRA, a low rank adapter on the diffusion model weight matrix for continual learning. They devise a regularization strategy for the introduced LoRA adapters and show that it is effective for learning multiple concepts. In addition to C-LoRA, the paper also proposes a customized token strategy by replacing object names with learned embeddings, to further reduce interference between concepts.

**Audience:**

Yes

**Broader Impact Concerns:**

This paper proposes methods to improve continual learning of multiple concepts in text-to-image generation models, which has potential negative ethical implications (e.g., deepfakes). These concerns are mostly discussed in the limitations section, but it may be helpful to include further discussion on the novel aspects of this work that are not present in prior papers (e.g., DreamBooth). In particular, I’d like to see some discussion on the implications of enabling generating realistic images with multiple concepts/people.

**Claims And Evidence:**

Yes

**Requested Changes:**

## Would strengthen the work:
- Human evaluations for comparing C-LoRA against existing baselines.
- Correlation between human judgement and the proposed MMD score metric.
- Qualitative results on multiple distinct concepts (person + landmark, or person + stuffed animal, etc.)

**Strengths And Weaknesses:**

## Strengths:
- The paper provides analysis on how existing customization methods (DreamBooth, Custom Diffusion) are unable to do the continual learning task, and often collapse to a single mode.
- In addition to performing well on continual customization of text-to-image diffusion, the paper also provides experimental results to show the effectiveness of C-LoRA on rehearsal-free learning for image classification, achieving SOTA (improving over approaches such as L2P, CODA-Prompt).


## Weaknesses:
- The paper proposes some new metrics for computing performance with different concepts (the MMD score). It will be very useful if the authors can also conduct some form of human evals, to ensure that the model indeed performs better than the baselines as evaluated by humans. The qualitative results presented in the paper seem to make this clear, but it will nevertheless be helpful to have a quantitative score that ascertains this.
- Most of the results seem to be on unimodal datasets (e.g., all faces, or all landmarks). Are there any results showing composition of multiple, distinct objects? For example, adding a unique stuffed toy as well as a person. Does the proposed method do better than baselines, or does C-LoRA mostly shine when the objects are of the same type?
- Missing references to earlier work on PEFT: (1) Houlsby, Neil, et al. "Parameter-efficient transfer learning for NLP." International Conference on Machine Learning. PMLR, 2019., (2) prefix tuning and prompt tuning papers. These references would be helpful to give a more comprehensive overview in section 2 of the history of PEFT.

---

> ### Author Response · Authors · 2024-01-29
> **Author Response to Reviewer Nanp**
>
> We are grateful to the reviewer for their constructive review. In our revised submission, we will incorporate the recommended references – thank you for highlighting their importance. We address the reviewer comments below:
>
> **On qualitative results on multiple distinct concepts**: We thank the reviewer for this suggestion and are in the process of generating results to fulfill this requirement. These results, along with a detailed analysis, will be included in both the revised manuscript and our subsequent author response.
>
> **On human evaluation**: While we acknowledge the reviewer's perspective on human evaluation, we maintain that the impact of catastrophic forgetting in competing methods is discernible through qualitative analysis and is pronounced in our current quantitative metrics. For instance, the effects of identity loss and image degradation are evident in Figure 1 of our paper. We believe that human evaluation holds greater relevance in research primarily focused on marginal improvements in image quality, as opposed to our work which concentrates on mitigating catastrophic forgetting. We also wish to highlight that additional metrics are provided in our appendix, including image alignment, text alignment, and the Kernel Inception Distance (KID) metrics, as reported in Kumari et al., 2022.
>
> **On broader impact concerns**: We agree that the concerns raised are crucial for understanding the broader impact of our work and its future iterations. In our revision, we will include an expanded discussion on these points. This will particularly focus on the ethical implications of generating realistic images featuring multiple concepts and people, with a special emphasis on the potential for disinformation and harm to reputations.

---

### Review · Reviewer_mJhg · 2024-01-23

**Summary Of Contributions:**

The paper proposes an extension of low-rank adaptation (LoRA) to the setting of continual learning for image generation within diffusion models. Applied to datasets of faces (CelebA-like) and landscapes/waterfalls, the authors train models on sequences of 10 concepts (10 identities of people, or "styles" of waterfalls). The model learns new LoRA parameters for each of these concepts, which have an additive effect on the training. The authors demonstrate improvements over various baseline models (e.g. LoRA, or training seperate adapters, or using other continual learning techniques like EWC) using their approach, and claim improved performance with a smaller memory overhead.

**Audience:**

Yes

**Broader Impact Concerns:**

The broader impact statement should be reworked, it is somewhat a bit informal. Please add a statement and references for the dataset, how the faces were selected, you may justify this with prior work etc. Provide more context like you already started in Appendix G. Justify why you chose a face dataset (vs. e.g., other objects) specifically, and touch on the implications this has for your work. Discuss how you picked the images. Write about the potential impact of generating new compositions of different persons, and put a disclaimer about how this work could and should not be used in production environments, and what the limitations are. What is your release policy on model weights, etc?

**Claims And Evidence:**

No

**Requested Changes:**

Please go through the list of weaknesses and comments I posted above. Some aspects are more minor, but a few would greatly improve the quality of the paper. The most important changes to make are:

- Run multiple dataset seeds, and avoid overfitting to a few selected concepts. Report mean and standard deviation (or s.e.m. in all tables)
- Evaluate what happens when more than 10 concepts are chosen (50, 100, etc), or justify why this is prohibitive (I do not really see why it would be)
- Clarify the memory overhead of the method, and how the weight matrices are stored exactly. Validate this by getting param counts from your model implementation, ideally.
- Rework the broader impact statement (see below)
- Clarify choice of baselines, clarify which is the "base algorithm" the method is added to, and consider to add additional ablations to motivate the design choices.
- broader impact statement:
  - "Specifically, we caution against training our method on large task sequences of 50 or 100 faces, as our approach may suffer". This was an immediate thought while reading the paper: I think you should include exactly this experiment. It seems clear that at some point, performance will degrade, but it would be interesting to get some insight in when this happens (maybe also depending on the size of the adaptation matrices, etc).

**Strengths And Weaknesses:**

## Strengths

The paper investigates an important and open problem in continual learning.

The considered continual adaptation setting -- to my knowledge -- extends previous studied settings (often classification tasks) on often smaller scale data (like MNIST, shuffled MNIST etc) to a generative computer vision problem on real images. This is an interesting contribution.

## Weaknesses

Overall, I find the idea and evaluation setting really interesting, and this could become a great TMLR paper. However, the execution and rigour in the evaluation is below TMLR standards. I would be happy to iterate with the authors on potential improvements of the paper.

Some questions to start with:

**Regarding memory:**

First, I am not fully clear on how the parameters are stored. In Eqs (2) and (3) there is a split between concepts 1...t-1 and the new matrices for task t. My guess is, you store the summed matrices for 1...t-1 and the individual matrices A and B for training, is that correct? For inference, you only need to store the overall summed matrix? In either case, I do not follow how you arrive at the parameter count in Table 1 and 2. The summed matrix does not decompose into invidiudal low-rank matrices any more, so this would imply storing the multiplied matrices (larger memory footprint). Could you confirm and clarify, I might be missing something.

Second, is there a difference in parameter count used during training and inference? If so, it would be fair to other methods to have two columns in Table 1 and 2 for this, and distinguish both cases. I think for example that it would be fine to have more parameters than e.g. standard LoRA, but this should be pointed out (if it is the case, the my point above).

Third, how are the parameters for Table 1 and 2 actually computed in practice? Do you get the numbers directly from your implementation (that would be the way), or is this an estimation not tied to your particular implementation? Just to double check the numbers, could you confirm that they actually correspond to something you can get out of your model/code?

Finally, is there memory overhead for integrating the additional tokens you need for your approach?

**Regarding hyperparameters:**

For hyperparameter tuning, I assume you sample a different set of concepts (identities) from the ones used for testing? Could you extend the appendix section on the tuning protocol, and potentially also add a few tables on hyperparameter variations and their impact on performance?

**Regarding depth of analysis**

- Why not plot performance of all methods across the 10 tasks? The temporal evolution of performance would be interesting to consider.
- What is the criterion for selecting the "concepts" (i.e., person identities) in particular? For running the experiments, how many different concepts do you run over?
- It would be interesting to see when the model reaches capacity

### Additional weaknesses/comments on the paper

These are chronologically, important points are marked with an, minor points are not marked.

- Is SOTA Custom Diffusion (Kumari et al., 2022) still the SOTA? Just checking, as the field moves fast -- was there no more recent work in 2023 to consider?
- "catastrophic forgetting": This is central topic of the paper, but the quantitative evaluation metrics do not touch on this sufficiently.
- abstract: "(i.e" seems to be a "e.g."
- "Continual Diffusion": I think this is not a good name, as it mixing the learning setup with an evaluation setup
- Some icons (like the lock) in Fig 3 are overlapping with text
- p.5 "respectfully" -> "respectively"
- Eq. 1
  - typeset the "diff", "forget" etc. correctly as text vs. math
  - please write out the individual components of the loss (Ldiff and Lforget)
- Eq. 2
  - Why not letting the sum run till t? Is this because the first matrix is stored as a single element in the model? This was not fully clear during reading, maybe clarify in the methods.
- "However, such a strategy stores a copy of original parameters, introducing significant overhead," -> can you quantify this overhead in terms of # of model params/relative increase in model params?
- Eq 3: Clarify that the | . | is the element wise absolute value (I suppose), and make clear that the norm is the Frobenius norm, i.e. change subscript from "2" to "F".
- Appendix C misses the search ranges for all hyperparameters
- "We highlight that the A and B are low-rank matrices and thus only incur small costs to training and storage": What is the exact memory footprint you incur during training and inference?
- Section 3.3
  - "Furthermore, we remove any object names from the input sequence to encourage unique personalized embeddings"
  - which object name? where are these listed? how were they picked
- Section 4
  - "However, using real images would require constant access to internet downloads, which would be highly impractical for our setting"
  - why exactly is this impractical?
  - I would prefer to have the metrics mentioned in the paper
  - typeset refs properly, "see Kumari et al. (Kumari et al., 2022) for more details" -> see \citet{...} for more details, or similar
  - "Baselines": Any new baselines that came out in 2023 and were dropped in the evaluation? If so, minimally cite them, and consider including the most recent SOTA model (if applicable)
- Section 4.1.
  - self-generated: Was the same self-generated data as in the referenced prior work used?
  - please cite numbers in sec 4.1. directly, and make statements about the improvements you want to highlight
  - why only 10 tasks? why not scaling to 100, or 1000s of tasks given that the data is self generated
- Table 1:
  - Bolding of 0.51 is off
  - please add error bars to the table. Ideally, run across different variations of your dataset (e.g., 3-5 repeats). If unfeasible, please let me know the compute budget as a justification for running only a single seed.
  - "Nparam" is misleading for a relative number. Also why not putting % in the table to make it more clear
  - why is the Nparam number the same as for LoRA? shouldnt you get a linear increase in memory with each added task? (same Q for Table 2). Also, dont you need to store the product of matrices vs. just two low rank matrices?
  - what is your baseline to study the effect of C LoRA specifically? i.e., which is the setting where the model etc. is exactly the same, just c lora is switched off? Do all methods use the exact same base model? if different models, what is the performance on the baseline task
  - remove the vertical lines in the tables / improve table typesetting
  - Why is the number of tasks in Table 1 not simply writen into the caption. What is the number of tasks in Table 2? (also seems to be 10?)
  - consider merging table 1 and 2, given that all is same except for numbers/datasets (could be arranged across columns), if this looks nicer in the paper
- In figure 4 and 5, was there a control experiment to make sure how the visual quality of the two methods matches in the non-continual setting? I.e., does the improved model performance stems from the method, or from a better baseline performance? or is that panel [b]?
- "requires that KV values of all tasks be stored indefinitely" -> where? and how can CD (merge) be worse than the LoRA approach presented here? Is this due to the regularizer? Is there evidence validating that the regularizer is indeed the cause for the increase in performance?
- Figure 6: I do not fully get the quadrant numbering. Is this done only for the figure, or also somehow for the model
- Section 5
  - ImageNet-R "because the distribution of training data has significant distance to the pre-training data (ImageNet), thus providing a fair and challenging problem setting." -> Please cite a reference
- " our best to engineer improved baselines": can you point to the table where u validate that the baselines are on par or beyond the original papers
- Table 3: Do you want to contextualize the ImageNet-R results also with ref numbers from imagenet models?

---

> ### Author Response · Authors · 2024-01-29
> **Author Response to Reviewer mJhg**
>
> We sincerely thank the reviewer for their thorough and detailed review. We appreciate the effort invested in suggesting comprehensive revisions to enhance the quality and rigor of our paper, potentially aligning it with TMLR standards.
>
> We acknowledge the extensive nature of the comments, which surpass the specific requested changes. To ensure efficient communication and timely revisions, we will address all minor concerns and questions raised by the reviewer in our revised paper. In this response, we will focus primarily on the requested changes and major clarifications.
>
> **On dataset seeds and hyperparameters**: We did not tune hyperparameters on the same task set as our evaluations. For our revised paper, we will extend the appendix section on the tuning protocol and add a few tables on hyperparameter variations. Moreover, we will convert our tables to include mean and standard deviation across multiple seeds. This will be posted after the full revisions are finished.
>
> **On beyond 10 concepts**: In our experiments, we observed a decrease in plasticity beyond 10 tasks, yet we maintained state-of-the-art performance. Furthermore, our limit exceeds the limit we found in Custom Diffusion, as demonstrated in Figure B of our Appendix. Our revision will explicitly include experiments and analyses demonstrating the capacity limits regarding the number of concepts.
>
> **On memory overhead**: The reviewer has raised an important clarifying point. We will briefly summarize it here, and revise the paper to include this crucial clarification. For our method, the number of parameters trained is 0.09% of the U-Net backbone (we normalize this number for easy comparison). In contrast, this number is 2.23% for Custom Diffusion. For **storage**, we have two options for our method: store the individual LoRA weights for each past concept or store the summation. For a 10-concept sequence, the storage cost would be 0.09% * 10 = 0.9%, which is less than Custom Diffusion. However, **our storage is bounded by the size of the summed multiplied matrices**, as pointed out by the reviewer. Thus, if the sum of individual LoRA weights were to exceed the size of the summed multiplied matrix, we would only store the summed multiplied matrix long term, and thus our max cost would be equivalent to the 2.23% of Custom Diffusion. **We will include the storage as an additional column in our results table in our revision.**
>
> We calculated these numbers using our code to ensure correction. Furthermore, the memory head of the tokens is negligible when reporting in percentage of the U-Net backbone, and is the same cost for all methods.
>
> **On broader impact**: Thank you for the comments; our revision will include an updated broader impact statement to address your concerns and comments.
>
> **On choice of baselines**: Custom Diffusion and DreamBooth were state-of-the-art and the most commonly used models at the time of our work; however, we understand the field moves fast with many follow-up papers. As mentioned in a comment to another reviewer, we are working to compare to a concurrent work posted to arXiv on November 22, 2023 (ZipLoRA). This will contextualize our method with the very latest state-of-the-art.
>
> The “base algorithm” for our method is Custom Diffusion (sequential), with LoRA (Sequential) representing a naïve combination of LoRA + Custom Diffusion. The rest of our contributions extend LoRA (Sequential). Full ablation results are available in Table C of our Appendix.

---

> > ### Author Response · Authors · 2024-01-29
> > **Author Response to Reviewer mJhg - Additional Clarifications**
> >
> > **Additional clarifications**:
> > - Our selection of identities for concepts was random.
> > - We did not plot results against the number of tasks due to computational costs and the complexity of adequately representing many metrics.
> > - Our F_mmd metric is specifically designed to quantitatively measure catastrophic forgetting.
> > - All methods use the same backbone and codebase. LoRA (sequential) represents the naïve combination of LoRA + custom diffusion, and serves as the main comparison where C-LoRA is “turned off”. Section D of our appendix provides direct analysis of LoRA vs C-LoRA, demonstrating that our contribution is crucial.
> > - Regarding equations 1 and 2, we will consider revising these if the reviewer insists, but currently prefer their existing format for generality and clarity.
> > - EWC requires a 200% increase in model parameters, as it requires storing 1) the current model parameters, 2) a copy of the old model parameters, and 3) the diagonal of the Fisher-Information matrix of the past model gradients with respect to the loss function.
> > - Regarding equation 3 – you are correct, and we will update the equation in the revised text.
> > - Regarding object names – following the style of Custom Diffusion, these object names are simply “person” and “waterfall”.
> > - For the baseline regularization used in Custom Diffusion, our implementations leveraged the self-generated data in the same way as Kumari et al., 2022 (using their code exactly). All methods in our results used the exact same regularization images for a fair comparison. This allows our method to work without a constant internet connection.
> > - We will run a small experiment to compare the methods in the offline setting to ensure a fair starting place.
> > - Regarding "requires that the KV values of all tasks be stored indefinitely”: we will clarify this text in our revision. In order to create “on the fly” merged models from CD (merge), one would require that all KV values of all tasks be stored indefinitely. Merging in the “continual manner”, as done in our paper, does not incur this large storage cost. We directly show in Appendix E that the performance of CD (merge) collapses after roughly 5-6 tasks. Specifically, CD (Merge) performs on par with our method until roughly 5-6 tasks, and then it severely degrades in performance. We note that our results demonstrate that LoRA + CD (sequential) performs better than CD (sequential), but emphasize that both of these methods severely underperform our C-LoRA.
> > - The quadrant numbering in Figure 6 is purely for matching target concepts with generated images.
> > - Regarding “our best to engineer improved baselines” – here, we are referring to our text-to-image experiments where zero continual learning baselines existed, motivating us to create EWC + LoRA and CD + LoRA as new comparisons to better contextualize our performance.
> > - In Table 3, we presented a very standard results table comparable to the CVPR 2023 paper whose code we used to generate our results and comparisons.

---

> > ### Comment · Reviewer_mJhg · 2024-01-29
> >
> > Dear authors, thanks a lot for the reply. Your revision plan sounds great, and I am looking forwards to the new experiments and addition to the tables. I think I understand the point about "On memory overhead" now, but will wait for the updates to the paper to double check. I would appreciate if you can use colors to highlight the differences between the revisions, if possible.
> >
> > Also, thanks for the note on baselines. To avoid false expectations, it would be perfectly fine for me and within the scope of TMLR if e.g. ZipLoRA outperforms yours on some of the metrics, or there is no clear winner --- what I care more about is a proper discussion/contextualization of the SOTA. So thanks for including this.

---

### Author Response · Authors · 2024-01-29
**Update 1 from Authors**

We kindly thank all of the reviewers for their constructive reviews. We are happy to see positive interest in our work on an “important and open problem in continual learning”, and furthermore we agree that the requested revisions will improve the quality of our work.

We have already provided preliminary responses to each reviewer, addressing numerous comments that did not necessitate additional experiments or results. We are currently working on addressing the remaining comments and fulfilling the requested revisions. A revised version of the paper, complete with detailed responses to all reviewer feedback, will be submitted shortly.

While we do not anticipate any responses from the reviewers to our initial replies until the full revision is completed, we felt it was important to address some of the feedback without delay. Once again, thank you for your valuable input and for guiding us towards improving our manuscript.

---

### Author Response · Authors · 2024-02-05
**Update 2 from Authors - Revision Uploaded**

We kindly thank all of the reviewers once more for their constructive reviews. We believe we have addressed all reviewer requests in our revised submission (revisions are marked in our document with both red text and red squares in the right margin).
- We have updated the parameter-efficient fine-tuning related works to add additional citations (Reviewer Nanp; Section 2 - Page 4)
- We have included qualitative results on multiple distinct concepts using the CustomConcept101 dataset (Reviewer Nanp; Appendix F - Page 20)
- We have updated the broader impact concerns (Section 7 - Page 12) to 1) better address the unique concerns of our continual setting (Reviewer Nanp + mJhg), 2) include additional info on the dataset (Reviewer mJhg), and 3) added a disclaimer on production environments  (Reviewer mJhg).
- We have extended the appendix section on the tuning protocol (Appendix C - Page 19)  and added a few plots (Appendix H - Page 21) on hyperparameter variations (Reviewer mJhg)
- We have extended our results to include mean and standard deviation over multiple seeds (Reviewer mJhg)
- We have included results and analysis on the capacity limits regarding greater than 10 concepts (Reviewer mJhg; Appendix I - Page 22)
- We have clarified the memory overhead details and included storage as an additional column in our results tables (Reviewer mJhg; Appendix C - Page 19)
- We have revised the text to clarify the storage of KV values in Custom Diffusion vs ours (Reviewer mJhg; Section 3.2 - Page 10 + Appendix C - Page 19)
- We have updated equation 3 to use the Frobenius norm  (Reviewer mJhg; Section 3.2 - Page 6)
- We have verified that our method is a fair starting place in offline experiments  (Reviewer mJhg; Appendix G - Page 21)
- We have included comparisons to a concurrent work (ZipLoRA) to contextualize our method with the very latest state-of-the-art (Reviewers mJhg + VgvP; Appendix A - Page 17); however, we note that our findings indicated  ZipLoRA does not perform as well as our C-LoRA, likely because ZipLoRA was originally conceived for concept/style integration, as opposed to learning a large number of subjects in a continual learning context

---

### Decision · Action_Editor_coUG · 2024-03-12

**Recommendation:** Accept with minor revision

**Comment:**

This paper proposes a method to allow text-to-image diffusion models to learn multiple concepts in a sequential manner. They do this effectively by introducing C-LoRA, a low rank adapter on the diffusion model weight matrix for continual learning. After author rebuttal, it received 3 Leaning Accept recommendations.

All the reviewers are happy about the paper after the revision, commenting that (1) C-LoRA is intuitively designed for continual multi-concept learning; (2) C-LoRA outperforms the previous multi-concept learning method, Custom Diffusion, and overall, this paper has made some interesting contributions to the community.

For the final version, there are some additional requests from Reviewer mJhg.

1. ZipLoRA should not be hidden in Tables A and B, it should be pulled into the main paper tables (Table A and B can stay additionally, of course). Likewise, the section about ZipLoRA in p.17 should go to the main paper/related work.

2. The adaptation plot in Figure E likewise seems important. It would be nice to minimally add the performance of other methods at x=10 Tasks. It would be even better if the authors had numbers for all methods across the 50 tasks, but I understand if this is prohibitive to run (but it would be a great resource to readers on where we are with continual learning in this setting). I would encourage the authors to minimally place this figure into the main paper, add the already run results into the figure, and add discussion on it as well, given that there is some space left in the main paper.

3. (Optional) As an additional suggestion which I would strongly recommend, but I will defer to the authors to decide, I would really encourage the authors to improve the plotting and typesetting quality of their figures and tables. The captions in e.g. Fig 4/5 could better match the paper font (or at least not being bold). For tables like Table 3, E, I would suggest to remove the vertical bars for clarity. For figures like Figure A, D, E, I would suggest re-plotting with proper fonts (e.g. Fig. E uses at least 4 different font sizes and styles) and sizes matching the paper.

Overall, the Action Editor would like to recommend Accept with Minor Revision, so that the above reviewer's request can be properly addressed.

**Audience:**

Yes

**Claims And Evidence:**

Yes

---

> ### Author Response · Authors · 2024-04-13
> **Response from Authors**
>
> We kindly thank the Action Editor and all reviewers for their constructive reviews and high-quality feedback. We have uploaded the final version of our paper, including the following requested minor revisions:
> 1. We have included the results on ZipLoRA in the main paper experiments. Furthermore, we have moved the discussion on ZipLoRA to the main paper related works and main paper experiments.
> 2. Thank you for this suggestion. We have moved the adaption plot and discussion in Figure E to the main paper. Given as this is a new experiment (i.e., x=10 is not equivalent to the settings in Tables 1 or 2), we ran new results to add other methods to the plot. We only included the most meaningful comparisons to retain readability.
> 3. We appreciate the optional suggestions on figures and tables. Our personal preference is to leave them as is. We did remove the vertical bars for Table 3 but left these bars in Table E to visually aid in separating the 3 different task sizes.